# mosGILT antibodies interfere with *Plasmodium* sporogony in *Anopheles gambiae*

Brady Dolan[1,4], Tomás Correa Gaviria [1,4], Yuemei Dong[2], Peter Cresswell[3], George Dimopoulos [2], Yu-Min Chuang [1] ✉ & Erol Fikrig [1] ✉

*Plasmodium*, the causative agents of malaria, are obtained by mosquitoes from an infected human. Following *Plasmodium* acquisition by *Anopheles gambiae*, mosquito gamma-interferon-inducible lysosomal thiol reductase (mosGILT) plays a critical role in its subsequent sporogony in the mosquito. A critical location for this development is the midgut, a tissue we show expresses mosGILT. Using membrane-feeding and murine infection models, we demonstrate that antibodies against mosGILT reduce the number of *P. falciparum* and *P. berghei* oocysts in the midgut and the infection prevalence of both species in the mosquito. mosGILT antibodies act in the mosquito midgut, specifically impacting the *Plasmodium* oocyst stage. Targeting mosGILT can therefore interfere with the *Plasmodium* life cycle in the mosquito and potentially serve as a transmission-blocking vaccine.

Malaria, caused by several *Plasmodium* species, is a major global health problem with over 200 million cases and 600,000 deaths in 2022[1]. Transmission of *Plasmodium* begins when mature male and female gametocytes within the peripheral blood of an infected human host are acquired by a mosquito during a blood meal. Following the blood meal, these gametocytes undergo differentiation into micro (male) and macro (female) gametes in the mosquito, initiating a process of fertilization that culminates in the formation of a zygote[2,3]. Zygotes then transform into ookinetes, which invade and traverse the midgut epithelium and become oocysts. Subsequently, mature oocysts rupture and release sporozoites into the hemolymph, which then infect the salivary glands, initiating a new infection when the mosquito takes a subsequent blood meal[4].

The emergence of drug-resistant protozoa and insecticide-resistant *Anopheles* mosquitoes highlights the need for additional strategies to help eliminate this disease[1]. Transmission-blocking interventions have gained prominence for their potential to disrupt the *Plasmodium* life stages associated with the mosquito. Transmission-blocking involves targeting *Plasmodium* at life stages between the gametocytes, that are ingested by the mosquito, and the sporozoites deposited into the vertebrate host by the mosquito biting. These life stages are of interest for transmission-blocking as they are where the *Plasmodium* parasite experiences major population bottlenecks. As an example, the mosquito only ingests a small number of gametocytes during a blood meal for as few as $10^3$ of the estimated $10^{10}$ *Plasmodium* gametocytes circulating in the peripheral blood of infected humans are ingested by an *Anopheles* mosquito[5–7]. In addition, only a few oocysts develop from hundreds of ookinetes in the mosquito midgut[8,9]. These observations underscore the midgut stage as an essential bottleneck of the *Plasmodium* life cycle, providing strategic insight for interventions in malaria control.

Transmission-blocking vaccines (TBVs) are a crucial component of transmission-blocking interventions. Parasite-based TBVs target proteins expressed by gametocytes, gametes, or the subsequent sporogonic stages of *Plasmodium*[10–13]. More recent developments involve targeting vector-based antigens to impede the penetration or development of *Plasmodium* in the mosquito midgut[2,14,15]. Overall, TBVs do not directly prevent infection in vaccinated individuals but

[1]Section of Infectious Diseases, Department of Internal Medicine, Yale University School of Medicine, New Haven, CT 06520, USA. [2]W. Harry Feinstone Department of Molecular Microbiology and Immunology, Bloomberg School of Public Health, Johns Hopkins University, Baltimore, MD 21205, USA. [3]Department of Immunobiology, Yale University School of Medicine, New Haven, CT 06520, USA. [4]These authors contributed equally: Brady Dolan, Tomás Correa Gaviria. ✉e-mail: yu-min.chuang@yale.edu; erol.fikrig@yale.edu

reduce the *Plasmodium* burden in mosquitoes and, therefore, indirectly offer protection to the community.

Modeling studies have predicted the utility of TBVs using data collected in a range of transmission settings across sub-Saharan Africa[16]. Research suggests that a TBV causing a 50% decrease in the mean *Plasmodium* infection prevalence in mosquitoes in a laboratory setting could result in a 72% reduction in malaria disease prevalence[17,18]. Additionally, another study evaluated the potential impact of a monoclonal antibody (TB31F) that targets Pfs48/45 on the surface of *Plasmodium* gametes within the mosquito midgut[19]. Modeling for this monoclonal predicted it could achieve an 80% decrease in the mean number of oocysts within mosquitoes within 5 months[19]. When administered alongside insecticide-treated nets and seasonal malaria chemoprevention, a single annual administration of TB31F could lead to a 54% and 74% reduction in malaria incidence in high and low-transmission settings, respectively[19–21]. These modeling studies demonstrate the potential role of TBVs as part of integrated malaria control strategies.

The mosquito gamma-interferon-inducible lysosomal thiol reductase (mosGILT) plays a crucial role in the reproductive fitness of *A. gambiae*[22]. CRISPR-introduced mosaic mutations in *mosGILT* resulted in impaired ovarian development in *A. gambiae*, with decreased production of 20-hydroxyecdysone and vitellogenin[22]. These *mosGILT* mutant mosquitoes also exhibited refractoriness to *Plasmodium* infection, which was associated with increased levels of thioester-containing protein 1 (TEP1)[22]. TEP1 is a mosquito immune effector critical for *Plasmodium* lysis within the vector[23–26]. Indeed, silencing *TEP1* in the *mosGILT* mutant mosquitoes rescued the development of the oocyst stage[22], providing a crucial link between *Plasmodium* sporogony in *A. gambiae* and the mosGILT protein. We, therefore, examined the capacity of mosGILT antibodies to influence the early stages of *Plasmodium* development within *A. gambiae* and thereby serve as a candidate for TBV.

## Results

### mosGILT is expressed in the mosquito midgut and upregulated during a blood meal

mosGILT plays a significant role in immune responses and *Plasmodium* development in *A. gambiae*[22,27]. Therefore, the impact of blocking mosGILT on these processes was investigated. Since the mosquito midgut is a critical location for *Plasmodium* development, the presence of mosGILT protein in the midgut was confirmed by immunoblot, using salivary glands as a positive control due to the high expression of mosGILT in this tissue (Fig. 1a). The presence of mosGILT in the mosquito midgut was further examined by confocal microscopy and detected in both the anterior and posterior regions of the midgut (Fig. 1b). The expression level of mosGILT in the midguts of blood-fed and sugar-fed mosquitoes was then determined using RT-qPCR (primers listed in Table S1). Significant upregulation of mosGILT was observed 3 and 5 days after blood-feeding (Mann–Whitney test, ** $p < 0.01$ and **** $p < 0.0001$, respectively), compared to sugar-feeding, with expression returning to baseline levels by day 7 (Fig. 1c).

### mosGILT antibodies prevent *Plasmodium* infection of *A. gambiae* in a membrane-based blood feeding model

Since mosGILT is present and upregulated in the midgut after blood feeding, we determined whether mosGILT antibodies could interfere with *Plasmodium* sporogony in the midgut of *A. gambiae*. To explore this, mosquitoes were examined after engorgement on *Plasmodium*-infected blood using a membrane-feeding model. A membrane-blood feeding model allows for the exposure of the mosquitoes to a defined number of gametocytes and a specific amount of antibodies. Purified rabbit mosGILT antibodies were mixed with *Plasmodium falciparum* gametocyte-supplemented blood with low or high gametocytemia. The control group received an equivalent amount of purified OVA

antibodies. In the low gametocytemia infection condition, the mosGILT antibody group showed a 43% reduction (Mann–Whitney test, ***, $p < 0.001$) in mean oocyst load from 6.0 to 3.4 (Fig. 2a). In addition, the infection prevalence in the mosGILT antibody group decreased by 29% (Fisher's exact test, ***, $p < 0.001$) compared to the control group (Fig. 2a). Even in the high gametocytemia infection assay at 7 days post-infection, the group treated with mosGILT antibody exhibited a 36% reduction in mean oocyst load from 32.8 to 21.1 (Mann–Whitney test, **, $p < 0.01$; Fig. 2b) and a 12% decrease in the infection prevalence compared to the control group (Fisher's exact test, *, $p < 0.05$; Fig. 2b). A rough transmission-blocking activity comparison of these mosGILT antibodies to a monoclonal antibody (mAb) against *P. falciparum* protein Pfs25 was also completed (Fig. S7). Significant reductions in *P. falciparum* oocyst infection intensity and infection prevalence were observed for both interventions, though, as expected, the Pfs25 mAb provided superior protection to the mosGILT antibodies.

These investigations were then extended to *P. berghei*, a *Plasmodium* species commonly used in small animal malaria models. Membrane-feeding assays using *P. berghei*-infected mouse blood with purified rabbit mosGILT antibodies were performed. The mosGILT antibody group exhibited an 82% reduction in mean oocyst load from 1.1 to 0.2 (Mann–Whitney test, ****, $p < 0.0001$; Fig. 2c), and the infection prevalence in the mosGILT antibody group decreased by 66% compared to the control group (Fisher's exact test, ****, $p < 0.0001$; Fig. 2c).

### mosGILT antibodies reduce the *Plasmodium* oocyst cross-sectional area

To examine if remaining oocysts in these experiments were impacted by the mosGILT antibody, the cross-sectional area of oocysts was examined. A 78% and 27% reduction in *P. falciparum* (Mann–Whitney test, ****, $p < 0.0001$, Fig. 3b) and *P. berghei* (Mann–Whitney test, *, $p < 0.05$, Fig. 3b) oocyst cross-sectional area, respectively, were observed in mosquitoes that fed on blood containing mosGILT antibodies compared to the control group.

### mosGILT antibodies reduce oocyst number and infection prevalence of *P. berghei* within *A. gambiae* using an in vivo animal model

The transmission-blocking activity of mosGILT antibodies was extended to an in vivo rodent model. Mice were first infected with *P. berghei*. Mice with comparable levels of parasitemia ( ~ 2% parasitemia) were used in these transmission-blocking studies. Mice were injected with mosGILT or control antisera 24 h prior to mosquito feeding. Mosquitoes that engorged on the animals given mosGILT antisera exhibited an 87% reduction in mean oocyst load from 3.0 to 0.4 (Mann–Whitney test, ****, $p < 0.0001$; Fig. 4a) and a 76% decrease in infection prevalence compared to the control (Fisher's exact test, ****, $p < 0.0001$; Fig. 4a).

To eliminate any potential effects of variable parasitemia between the control and mosGILT antisera-injected mice, the model was modified to use the same *P. berghei*-infected mouse. Mosquitoes were first allowed to feed on an individual *P. berghei*-infected mouse. These mosquitoes were considered the control group. Immediately after feeding, the infected mouse was administered purified rabbit mosGILT antibodies via retro-orbital injection. One hour later, a second group of mosquitoes was provided a blood meal from the same animal, and these mosquitoes were considered the intervention group. The mosGILT antibody group exhibited a 62% reduction in mean oocyst load from 35.3 to 13.3 (Mann–Whitney test, **, $p < 0.01$; Fig. 4b) and a 26% decrease in infection prevalence compared to the control group (Fisher's exact test, *, $p < 0.05$; Fig. 4b).

This one-mouse in vivo study was also completed with a mosGILT monoclonal antibody that was generated. *A. gambiae* that fed on the mosGILT monoclonal showed an 82% reduction in mean oocyst load

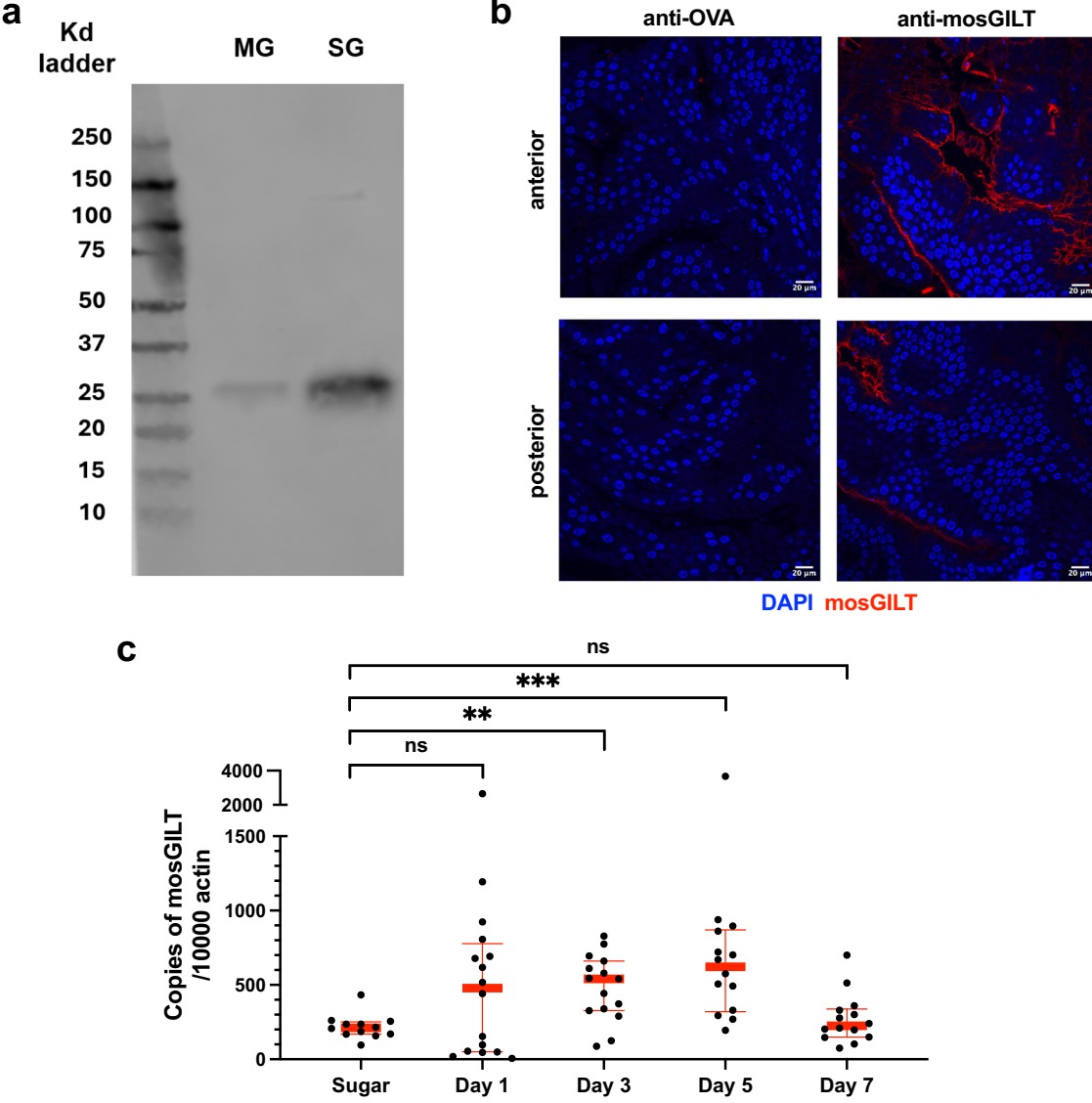

**Fig. 1 | mosGILT is expressed in the mosquito midgut and upregulated in response to a blood meal. a** Immunoblots of *A. gambiae* midgut (MG) and salivary glands (SG) probed with rabbit mosGILT polyclonal antibodies at a 1:10000 dilution. **b** Immunostaining of the anterior (upper panels) and posterior (lower panels) MG regions using purified mosGILT mouse polyclonal antibodies (0.8 μg/μl) at a 1:500 dilution (red). Alexa Fluor 647 goat anti-mouse IgG served as the secondary at a 1:1000 dilution. Blue: DAPI staining of the nuclei. Ovalbumin (OVA) purified mouse polyclonal antibody served as a negative control. **c** RT-qPCR analysis of mosGILT expression levels in midguts from sugar-fed mosquitoes (control, $n = 12$) and midguts collected 1 ($p = 0.5369$, $n = 16$), 3 ($p = 0.0014$, $n = 15$), 5 ($n = 14$), and 7 ($p = 0.5604$, $n = 14$) days post-blood meal. Each dot represents an individual mosquito midgut, with the thick horizontal lines indicating medians and thin lines denoting upper and lower quartiles. Two experimental replicates were performed. Two-Tailed Mann-Whitney was used to determine significance (*ns*: insignificant, **$p \leq 0.01$, ****$p \leq 0.0001$). Source data are provided as a Source Data file.

from 1.7 to 0.3 (Mann–Whitney test, ****, $p < 0.0001$; Fig. 4c) and a 75% decrease in the infection prevalence compared to the control group (Fisher's exact test, ****, $p < 0.0001$; Fig. 4c).

### Active immunization with mosGILT has transmission-blocking activity against *P. berghei* in *A. gambiae* mosquitoes

To assess the potential of mosGILT as a malaria transmission-blocking vaccine (TBV) target, a proof-of-concept study was conducted using an in vivo rodent model through active immunization of mice with recombinant mosGILT. Groups of C57BL/6 female mice were primed with mosGILT or control (OVA) protein and boosted twice at two-week intervals, as illustrated in Fig. 5a. Mice with strong anti-mosGILT antibody titers were then infected with *P. berghei*-infected red blood cells (Fig. 5b). Mosquitoes were allowed to take a blood meal on the *P.*

*berghei*-infected mice. The mosGILT-immunized group exhibited an 80% reduction in mean oocyst load from 36.5 to 7.5 (Mann–Whitney test, ****, $p < 0.0001$; Fig. 5c) and a 53% decrease in infection prevalence compared to the control group (Fisher's exact test, ****, $p < 0.0001$; Fig. 5c).

### mosGILT antibodies act on the oocyst stage of *P. berghei* within *A. gambiae*

To assess the infection stage-specificity of mosGILT antibody-mediated *Plasmodium*-blocking, mosquitoes were fed on mice administered mosGILT or control antisera. After 18 h, mosquito midguts were collected to determine the total number of ookinetes in the blood bolus and those traversing the midgut epithelium. No difference in the total number of ookinetes per midgut was observed between the

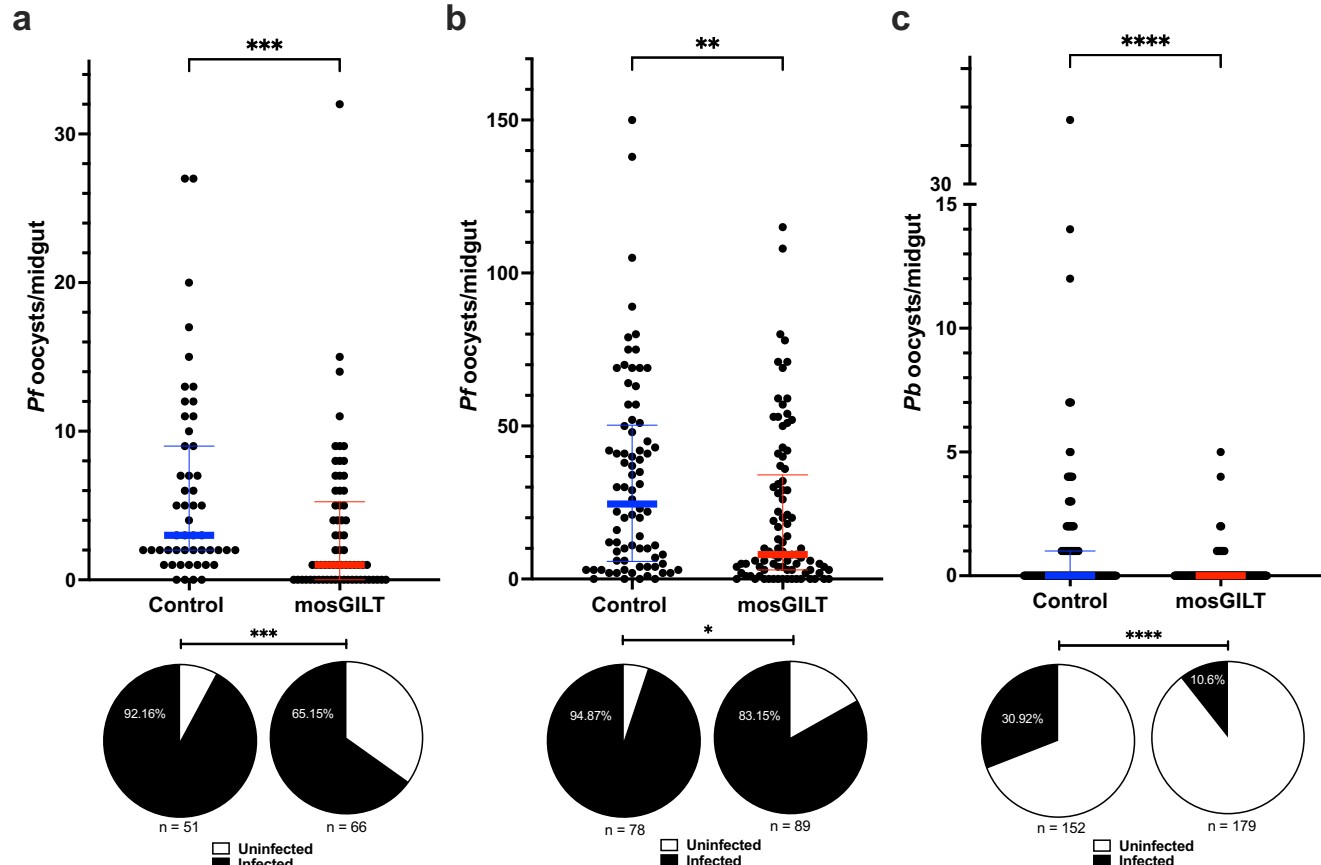

**Fig. 2 | mosGILT antibodies decrease *P. falciparum* and *P. berghei* oocyst infection prevalence in a membrane-feeding model. a, b** Dot-plots showing the number of *P. falciparum* oocysts per midgut (**a**, *p* = 0.0009; b, *p* = 0.0007) and infection prevalence (pie charts) (**a**, *p* = 0.0048; **b**, *p* = 0.0260) from mosquitoes that took a blood meal with a low gametocytemia blood (**a**, 0.008%) and high gametocytemia blood (**b**, 0.5%) with a final concentration of 0.2 mg/ml mosGILT (**a**, *n* = 66; **b**, *n* = 89) or control (**a**, *n* = 51; **b**, *n* = 78) antibodies. **c** Dot-plot of *P. berghei* oocysts counts per midgut (*p* = 0.0157) and infection prevalence (pie charts) from mosquitoes that took a blood meal from the feeder with 0.2 mg/ml mosGILT (*n* = 179) or control (*n* = 152) antibodies. Oocysts were counted seven days

post-infected blood meal. Each dot represents the number of oocysts in an individual mosquito midgut. The blue and red horizontal lines indicate the medians for the control and intervention groups, respectively. The thin lines indicate upper and lower quartiles. Low and high *P. falciparum* experiments were replicated a total of three times. *P. berghei* experiment was replicated more than three times. Two-Tailed Mann-Whitney was used to determine the significance of oocyst abundance. Fisher's exact test was used to compare infection prevalence values (**p* ≤ 0.05, ***p* ≤ 0.01, ****p* ≤ 0.001, *****p* ≤ 0.0001). Source data are provided as a Source Data file.

mosquitoes that fed on *P. berghei*-infected mice given mosGILT or control antisera (Fig. S1). The lack of impact on the abundance of ookinetes in the midgut shows that the mosGILT antibodies do not target the earlier stages of the *Plasmodium* life cycle, including gametocyte, gamete, and zygote. This finding suggests that the mosGILT antibodies impact ookinete invasion of the midgut epithelium or the oocyst stage of the *Plasmodium* life cycle.

To further elucidate the *Plasmodium* stage impacted by mosGILT antibodies, an experiment was performed in which the mosquitoes were first given a partial blood meal from a *P. berghei*-infected mouse, followed by a complete blood meal 48 h later from a mouse that had been passively immunized with mosGILT or control antisera. (Fig. 6a, b). Ingestion of mosGILT antisera 48 h after a partial blood meal from a *P. berghei*-infected mouse led to a 69% reduction in mean oocyst load from 10.6 to 3.3 (Mann–Whitney test, ***, *p* < 0.0001; Fig. 6c) and 31% reduction in infection prevalence compared to the control (Mann–Whitney test, **, *p* < 0.0001; Fig. 6c). Since ookinetes invade the midgut epithelium within 18–36 h after blood feeding, these results suggest that mosGILT antibodies most likely impact the oocyst stage of *Plasmodium* development, specifically between 2 and 7 days post-infected blood meal.

## mosGILT antibodies mediate transmission-blocking activity in the midgut and not the hemolymph

Next, an assay was performed where mosGILT antibodies were provided orally or via intrathoracic injection 48 h before feeding on a *P. berghei-infected* mouse to assess whether the effect occurs in the midgut (oral) or hemolymph (intrathoracic). For the feeding-based assays, a variation of the partial feeding scheme was utilized; here the antibody was ingested orally prior to *Plasmodium* (Fig. 7a). Ingestion of mosGILT antisera 48 h before the mosquitoes fed on *P. berghei*-infected blood meal resulted in a 73% reduction in mean oocyst load from 17.5 to 4.8 (Mann–Whitney test, **, *p* < 0.01; Fig. 7b). No difference in infection prevalence was observed.

Injected mosquitoes were held for 48 h, following which each group was allowed to feed until fully engorged on a *P. berghei*-infected mouse. Oocysts were counted eight days after the infected blood meal. Injection of purified mosGILT antibodies into the hemolymph 48 h before a *P. berghei*-infected blood meal did not alter the *P. berghei* oocyst burden when compared to control mosquitoes inoculated with control antibody (Mann–Whitney test, ns, *p* > 0.05; Fig. 7c). The lack of transmission-blocking activity when the mosGILT antibodies were administered into the hemolymph, at a concentration likely higher than what was consumed orally, indicates that the mosGILT antibodies

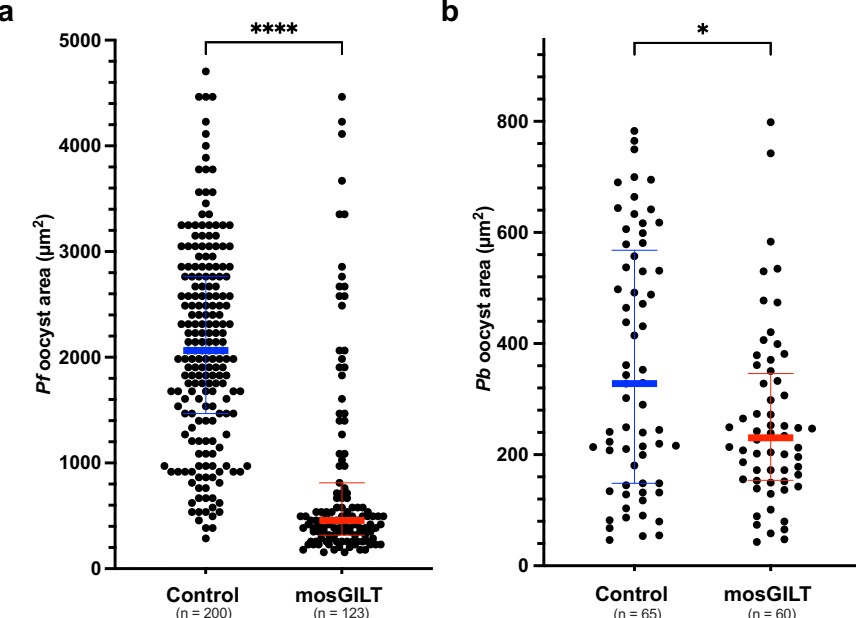

**Fig. 3 | mosGILT antibodies decrease *P. falciparum* and *P. berghei* oocyst cross-sectional area.** Cross-sectional area (size) of oocysts from mosquitoes that took a *Plasmodium falciparum* (**a**, *Pf*) or *Plasmodium berghei* (**b**, *Pb*, p = 0.0308) infected blood meal from a membrane feeder with mosGILT (**a**, n = 123; **b**, n = 60) or control (**a**, n = 200; **b**, n = 65) antibodies. Oocysts were imaged seven days post-infectious blood meal. Cross-sectional area was measured using ImageJ. Each dot represents the cross-sectional area of an individual oocyst. The blue and red horizontal lines indicate the medians for the control and intervention groups, respectively. The thin lines indicate upper and lower quartiles. *P. falciparum* and *P. berghei* experiments were replicated a total of three times. Two-Tailed Mann-Whitney was used to determine the significance of oocyst abundance (*$p \leq 0.05$, ****$p \leq 0.0001$). Source data are provided as a Source Data file.

may need to be introduced into the midgut through a blood meal to be effective.

## mosGILT antibodies decrease the expression of the *ecdysone receptor* and *lipophorin* in the *A. gambiae* midgut

mosGILT knockout mutant mosquitoes exhibit impaired ovarian development and decreased *Plasmodium* infection[22]. To assess whether mosGILT antibodies influence *A. gambiae* reproductive fitness, mosquitoes that fed on mice administered mosGILT or control antisera were examined. In contrast to mosGILT mutants with germline alterations, ovarian development (Fig. S2), egg-laying capacity (Fig. S3a), and larval hatch rate (Fig. S3b) were not impaired in mosquitoes that ingested mosGILT antibodies, compared to controls. These findings align with observations in adult mosquitoes where *mosGILT* was silenced via RNA interference with ds*RNA*[22].

The mosGILT-dependent global transcriptional responses in *A. gambiae* were recently elucidated through mRNA seq analysis of *mosGILT* mutant mosquitoes[27]. mosGILT is primarily involved in nucleocytoplasmic transport, metabolism, reproduction, and immune responses. From the *mosGILT* mutant mosquito RNAseq data, the 32 most up-regulated and down-regulated genes (Table. S2) were selected for further evaluation in mosquitoes that fed on mice passively immunized with mosGILT or control antisera, as described earlier, using RT-qPCR. Among the 32 genes of interest (Fig. S4a–ae), only two were found to be significantly down-regulated: *ecdysone receptor* (*EcR*) (Mann–Whitney test, ***, $p < 0.001$, Fig. 7d) and *apolipophorin* (*Lp*) (Mann–Whitney test, *, $p < 0.05$, Fig. 7e). To further confirm these results and eliminate variability between mice, the same *P. berghei*-infected mouse was used to feed both groups, as described previously. Both *EcR* (Mann–Whitney test, **, $p < 0.01$, Fig. 7f) and *Lp* (Mann–Whitney test, *, $p < 0.05$, Fig. 7g) were significantly down-regulated, consistent with the previous experiment.

To correlate the transcriptional findings with the protein expression of *A. gambiae*, hemolymph droplets were collected from 15 mosquitoes at 2, 3, and 4 days after a blood meal on mice immunized with mosGILT or control (GST) antisera. Proteins in the hemolymph were separated by SDS-PAGE, stained with Coomassie blue, and quantified by densitometry using ImageJ (FIJI). Mosquitoes that engorged on animals administered mosGILT antisera exhibited a decrease in apolipophorin I protein level, with reductions of 15%, 16%, and 31% on the 2, 3, and 4 days after the blood meal, respectively, compared to the control group (Fig. S5). These findings suggest a potential association between the administration of mosGILT antibodies and a reduction in *Lp* protein levels in *A. gambiae*.

## Discussion

TBVs have been explored as an approach in the fight against malaria to mitigate problems caused by the emergence of insecticide-resistant mosquitoes and drug-resistant *Plasmodium*. These vaccines mainly target parasite antigens expressed during the sexual and sporogonic stages of the parasite[2,4,7], but targeting mosquito-derived antigens that play roles in vector-parasite interactions can also contribute to transmission-blocking[2]. Antibodies against *A. gambiae* aminopeptidase N (AgAPN1)[14,28–30], which is expressed in the mosquito midgut lumen, and the fibrinogen-related protein 1 (FREP1)[15,31–33], expressed in the peritrophic matrix of the mosquito midgut, also have the ability to reduce the *Plasmodium* burden within the mosquito.

mosGILT plays a role in various biological processes within *A. gambiae*, including metabolism, immunity, and by supporting *Plasmodium* sporogony[22,27]. As mosGILT is expressed in the midgut and upregulated following blood-feeding, the transmission-blocking potential of mosGILT antibodies was examined. A membrane-based blood-feeding system demonstrates that polyclonal antibodies targeting mosGILT significantly reduces the oocyst infection intensity and prevalence for both *P. falciparum* and *P. berghei*. To further assess mosGILT antibodies anti-*Plasmodium* activity in vivo, a *P. berghei*-infected mouse model was employed. Four separate interventions - mosGILT antisera, polyclonal antibodies, monoclonal antibodies, and active immunization - all caused a significant decrease in oocyst infection intensity and prevalence. Importantly, the range of mean

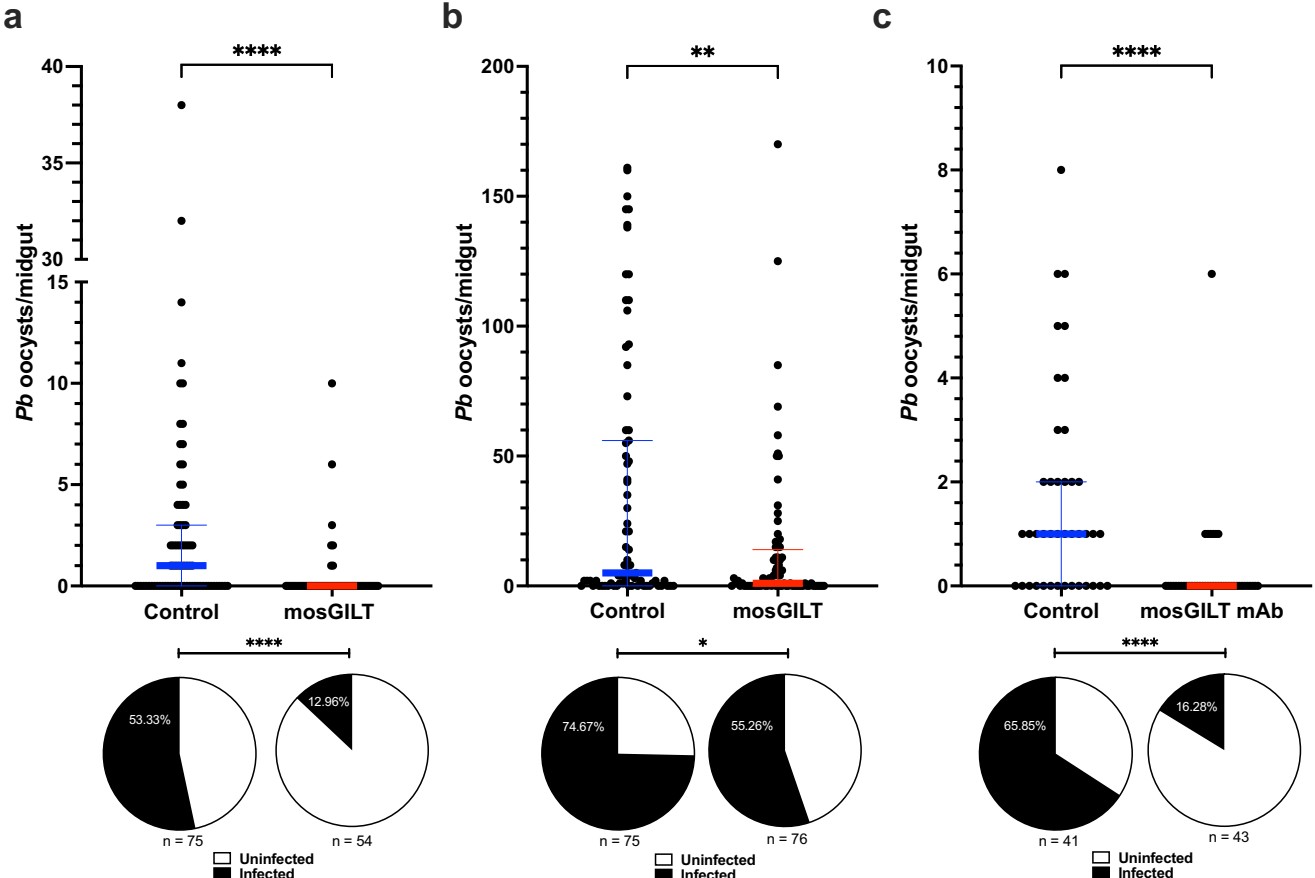

**Fig. 4 | mosGILT antibodies decrease *P. berghei* oocyst number and infection prevalence after passive immunization. a** Number of *P. berghei* oocysts per midgut and infection prevalence (pie charts) from mosquitoes that took a blood meal from a *P. berghei*-infected mouse passively immunized with mosGILT ($n = 54$) or control ($n = 75$) antisera. **b** Number of *P. berghei* oocysts per midgut ($p = 0.0025$) and infection prevalence (pie charts) ($p = 0.0168$) from mosquitoes that took a blood meal from a *P. berghei*-infected mouse before (control; $n = 75$) and after retro-orbital passive immunization with purified rabbit mosGILT polyclonal antibodies ($n = 76$). **c** Number of *P. berghei* oocysts per midgut and infection prevalence (pie charts) from mosquitoes that took a blood meal from a *P. berghei*-infected

mouse before (control; $n = 41$) and after retro-orbital immunization with mosGILT monoclonal antibodies (mosGILT mAb; $n = 43$). Oocysts were counted seven days post-infected blood meal. Each dot represents the number of oocysts in an individual mosquito midgut, and the blue and red horizontal lines indicate the medians for the control and intervention groups, respectively. The thin lines indicate upper and lower quartiles. At least two experimental replicates were performed. Two-Tailed Mann-Whitney was used to determine the significance of oocyst abundance. Fisher's exact test was used to compare infection prevalence values (*$p \leq 0.05$, **$p \leq 0.01$, ****$p \leq 0.0001$). Source data are provided as a Source Data file.

oocysts intensities observed across these various controls (1.1 – 36.5) enables the assessment of transmission-blocking activity under different conditions, including real-world mosquito infections, where mean parasite burdens are normally between 1 – 5 oocysts per midgut[20,34].

Overall, these data demonstrate that mosGILT has the potential to serve as a *Plasmodium* transmission-blocking target. Administering mosGILT antibodies through membrane feeding resulted in significant reductions in both oocyst intensity and overall infection prevalence. *P. falciparum* transmission-blocking activity has been demonstrated with targets such as Pfs25[35], Pfs230[35], FREP1 fibrinogen-like domain antibodies[15], and AgAPN1 antibodies under similar conditions[36]. Active immunization with recombinant mosGILT exhibited a similar activity to when mice were actively immunized with a Pfs25 DNA vaccine[37], suggesting that a combination of a mosquito target such as mosGILT and a *Plasmodium*-specific target such as Pfs25 would be a promising strategy. Future studies on mosGILT as a TBV antigen should first identify the most potent mosGILT epitope(s) and then compare their activity to existing targets under the same experimental conditions.

The specific stages of the sporogonic cycle where mosGILT antibodies have a transmission-blocking effect was examined. A combination of selective and partial feeding studies determined that the

number of oocysts was reduced—with no impact on the number of ookinetes. Therefore, these studies collectively demonstrate that mosGILT antibodies mainly affect the oocyst stage of *Plasmodium* sporogony. This effect aligns with the peak upregulation of mosGILT that was observed 3 to 5 days after a blood meal.

Studies have shown that antibodies ingested in a blood meal can pass into the hemolymph during digestion in various arthropods, including *Anopheles*[38-49]. Therefore, whether mosGILT antibodies act in the midgut or the hemocoel was examined. Direct injection of mosGILT antibodies into the hemocoel, bypassing the midgut basal lamina, did not influence *Plasmodium* development. This further confirms that the primary site of action for mosGILT antibodies is in the mosquito midgut. It is also possible that the mosGILT antibody has lower stability in the hemolymph, or that intrathoracic injection may have induced some degree of immune priming in the control mosquitoes, resulting in lower parasite intensity and, consequently, no additional effect from the mosGILT antibody.

To examine the potential mechanism of action of how mosGILT antibodies impact the number of oocysts, RNAseq data of differentially regulated genes in mosGILT knock-out (KO) *A. gambiae* compared to control mosquitoes was examined[27]. A subset of the most up-regulated and down-regulated genes in the mosGILT KO mosquitoes was

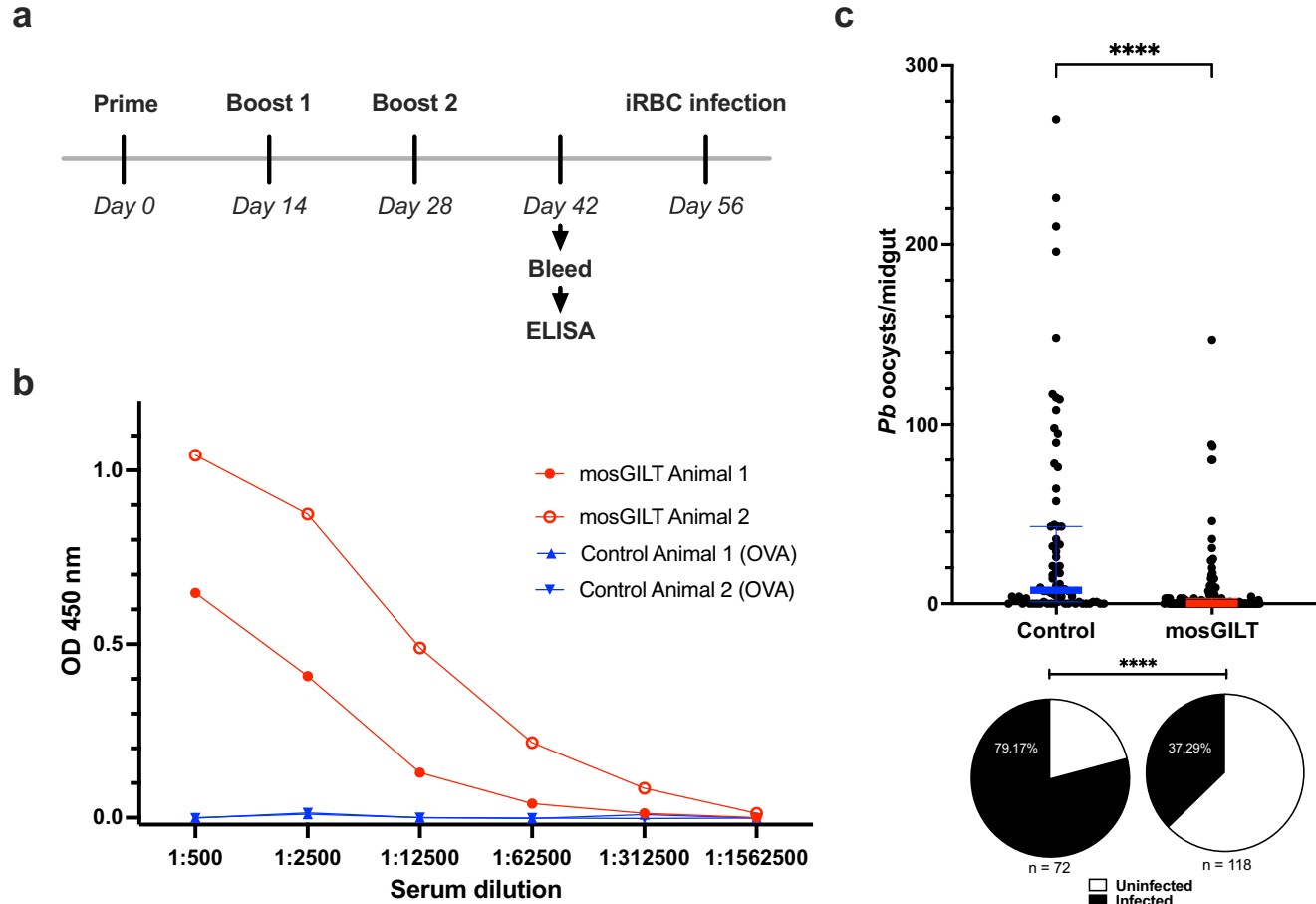

**Fig. 5 | Active immunization with mosGILT decreases *P. berghei* oocyst number and infection prevalence. a** Experiment scheme for C57BL/6 female mice injected with recombinant mosGILT (mosGILT) or control protein (OVA), and boosted twice, at two-week intervals. **b** Two weeks after the final boost, blood samples were taken from the mice. Sera were examined for mosGILT-specific antibodies by ELISA. Mice with strong titers were challenged with *P. berghei*-infected red blood cells (iRBC). Mosquitoes blood fed on the mice 5 or 6 days after *Pb* infection. **c** Number of *P. berghei* oocysts per midgut and infection prevalence (pie charts) of mosquitoes that took a blood meal from a *P. berghei*-infected mouse actively immunized with mosGILT ($n = 118$) or control ($n = 72$) protein. Oocysts were counted seven days post-infected blood meal. Each dot represents the number of oocysts in an individual mosquito midgut, and the blue and red horizontal lines indicate the medians for the control and intervention groups, respectively. The thin lines indicate upper and lower quartiles. Three experimental replicates were performed. Two-Tailed Mann-Whitney was used to determine the significance of oocyst abundance. Fisher's exact test was used to compare infection prevalence values (****$p \leq 0.0001$). Source data are provided as a Source Data file.

compiled (Table S1). As expected, these selected genes played roles in immune responses, metabolism, and reproduction[27]. The expression levels of this compiled list of genes were compared between mosquitoes given mosGILT and control antibodies via RT-qPCR. Analysis revealed significant downregulation of *EcR* and *Lp*. Notably, apolipophorin I protein levels exhibited reductions of 15%, 16%, and 31% on the 2, 3, and 4 days after the administration of mosGILT antibodies, respectively. Indeed, a study by Werling et al. revealed that silencing of *Lp* or *EcR* leads to a significant decrease in early oocysts[50]. Interestingly, mosquitoes in which *EcR* has been silenced showed an increased expression of *Lp* and a higher lipid content in their midguts after a blood meal, which led to an accelerated growth of the remaining oocysts and a shortened extrinsic incubation period. When *EcR* was silenced with *Lp*, there was a significant decrease in both oocyst number and size, suggesting that the role of *Lp* in lipid trafficking is crucial for oocyst development[50]. These findings suggest a potential association between the administration of mosGILT antibodies, reduced *Lp* levels, and decreased oocyst number and size. Future studies can elucidate the mechanisms through which mosGILT antibody administration elicits decreases in Lp and EcR levels.

As *Plasmodium* develops within the mosquito midgut, interactions with mosquito proteins play a crucial role in supporting its development, leading to a successful mosquito infection. This data collectively demonstrates that mosGILT antibodies can disrupt oocyst development, thereby decreasing the infection prevalence within the mosquito vector. Since mosGILT shares only 19% protein homology with human GILT, it is unlikely that cross-reactive antibodies would be generated upon human immunization with mosGILT. However, examining the efficacy of a mosGILT epitope that lacks any similarity to human GILT can also be assessed. In the landscape of TBVs, current clinical trials mainly focused on *P. falciparum* antigens such as Pfs230[51–56], Pfs48/45[50,56–60], expressed in gametocytes and gametes, as well as Pfs25[51,61,62], expressed on the surface of zygotes/ookinetes. Notably, a combination of vaccine antigens targeting both *Plasmodium* and mosquito proteins could act synergistically, enhancing the transmission-blocking efficacy and potentially enabling the vaccines to inhibit malaria transmission completely.

## Methods
### Ethics statement
Mice were housed at the Yale Animal Resource Center at Yale University and handled under the Guide for the Care and Use of Laboratory Animals of the National Institutes of Health. The animal experimental protocol was approved by the Institutional Animal Care

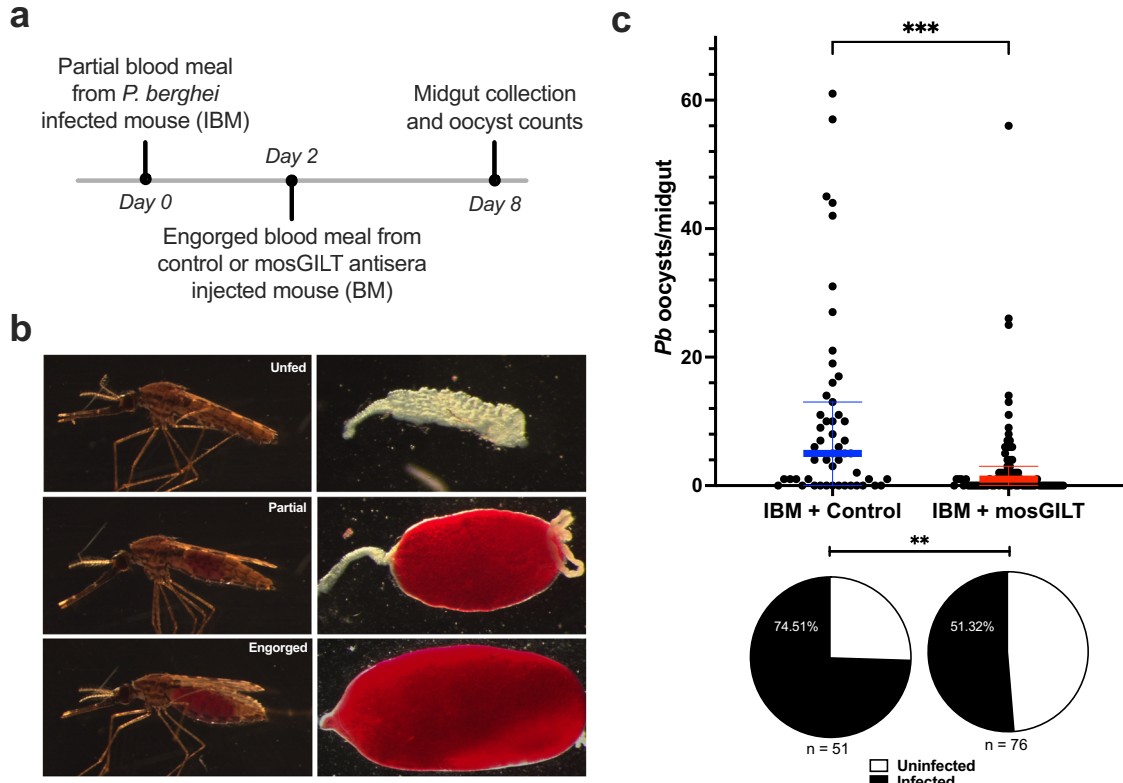

**Fig. 6 | mosGILT antibodies reduce the sporogony of *P. berghei* in *A. gambiae* by acting on the oocyst stage of the parasite. a** Illustration of the experimental scheme of a partially infected blood meal (IBM) followed by an antisera blood meal. **b** Visual representations of mosquitoes that had a complete blood meal (Engorged), partial blood meal (Partial), or no blood meal (Unfed). Mosquitoes that visually matched the partial blood meal status were used in this experimental scheme. **c** Number of *P. berghei* oocysts per midgut (*p* = 0.0001) and infection prevalence (pie charts) (*p* = 0.0099) of mosquitoes that took a partial IBM followed by a blood meal 48 h later from a GST (IBM + Control; *n* = 51) or mosGILT antisera

(IBM + mosGILT; *n* = 76) immunized mouse. Oocysts were counted eight days post-IBM. Each dot represents the number of oocysts in an individual mosquito midgut, and the blue and red horizontal lines indicate the medians for the control and intervention groups, respectively. The thin lines indicate upper and lower quartiles. Two experimental replicates were performed. Two-Tailed Mann-Whitney was used to determine the significance of oocyst abundance. Fisher's exact test was used to compare infection prevalence values (**$p \leq 0.01$, ***$p \leq 0.001$). Source data are provided as a Source Data file.

and Use Committee of Yale University (protocol permit no. 2023–07941). Human blood for *P. falciparum* cultures and mosquito infections was collected from a pool of pre-screened donors under an institutional review board–approved protocol at Johns Hopkins University (protocol NA00019050).

## Animals

*A. gambiae* (4arr strain, MRA-121, MR4, ATCC; Keele strain, Johns Hopkins Malaria Research Institute) mosquitoes were raised at 27 °C, 80% humidity, under a 12/12-hour light/dark cycle and maintained with 10% sucrose under standard laboratory conditions. Swiss Webster and C57BL/6 (6–8-week-old females) were purchased from Charles River Laboratories (Wilmington, MA) and housed under a 12/12-hour light/dark cycle with temperature being maintained at 20–26 °C.

## Antibody blocking assays of *P. falciparum* infection in *A. gambiae*

To determine the transmission-blocking activity of mosGILT antibodies, female *A. gambiae* were fed on NF54 *P. falciparum* gametocyte cultures (provided by the Johns Hopkins Malaria Institute Core Facility; NF54 obtained from MR4) mixed with purified rabbit mosGILT polyclonal IgG antibodies (intervention group) or rabbit ovalbumin (OVA) polyclonal antibodies (control group), at a final concentration of 0.2 mg/ml, in two settings, one with low gametocytemia (0.08%), and the other with high gametocytemia (0.5%), through artificial membranes at 37 °C[63–65]. The adult mosquitoes were starved for 3 to 5 h

before feeding to ensure engorgement. Unfed mosquitoes were removed immediately after blood feeding, and the remaining mosquitoes were incubated at 27 °C maintained on 10% sucrose solution. Seven days post-infection mosquito midguts were collected, and the number of oocysts was determined using a phase-contrast microscope (Leica). Low and high *P. falciparum* experiments were replicated a total of three times.

## Antibody blocking assays of *P. berghei* infection in *A. gambiae*

*P. berghei* (ANKA GFPcon 259cl2, MRA-865, ATCC) was maintained by serial passage in an 8-week-old female Swiss Webster as described previously[66,67]. Briefly, eight-week-old female Swiss Webster mice were challenged with *P. berghei*-infected red blood cells by intraperitoneal injection and used for the following experiments:

## Artificial membrane feeding studies

Mosquitoes were starved for at least 18 h before exposure to infected blood collected from infected Swiss Webster mice that presented with ~2% parasitemia. Murine parasitemia was monitored by Giemsa stain of blood smears. Blood was treated with heparin to prevent coagulation and mixed with purified rabbit mosGILT or OVA antibodies at a concentration of 0.2 mg/ml and immediately transferred into a Hemotek blood-feeding system with a hog intestine membrane. After 1 h of blood feeding, engorged mosquitoes were sorted, kept in paper cups covered with a net, and transferred to a chamber set to 20 °C. Mosquitoes were maintained with a cotton wick soaked in 10% sucrose

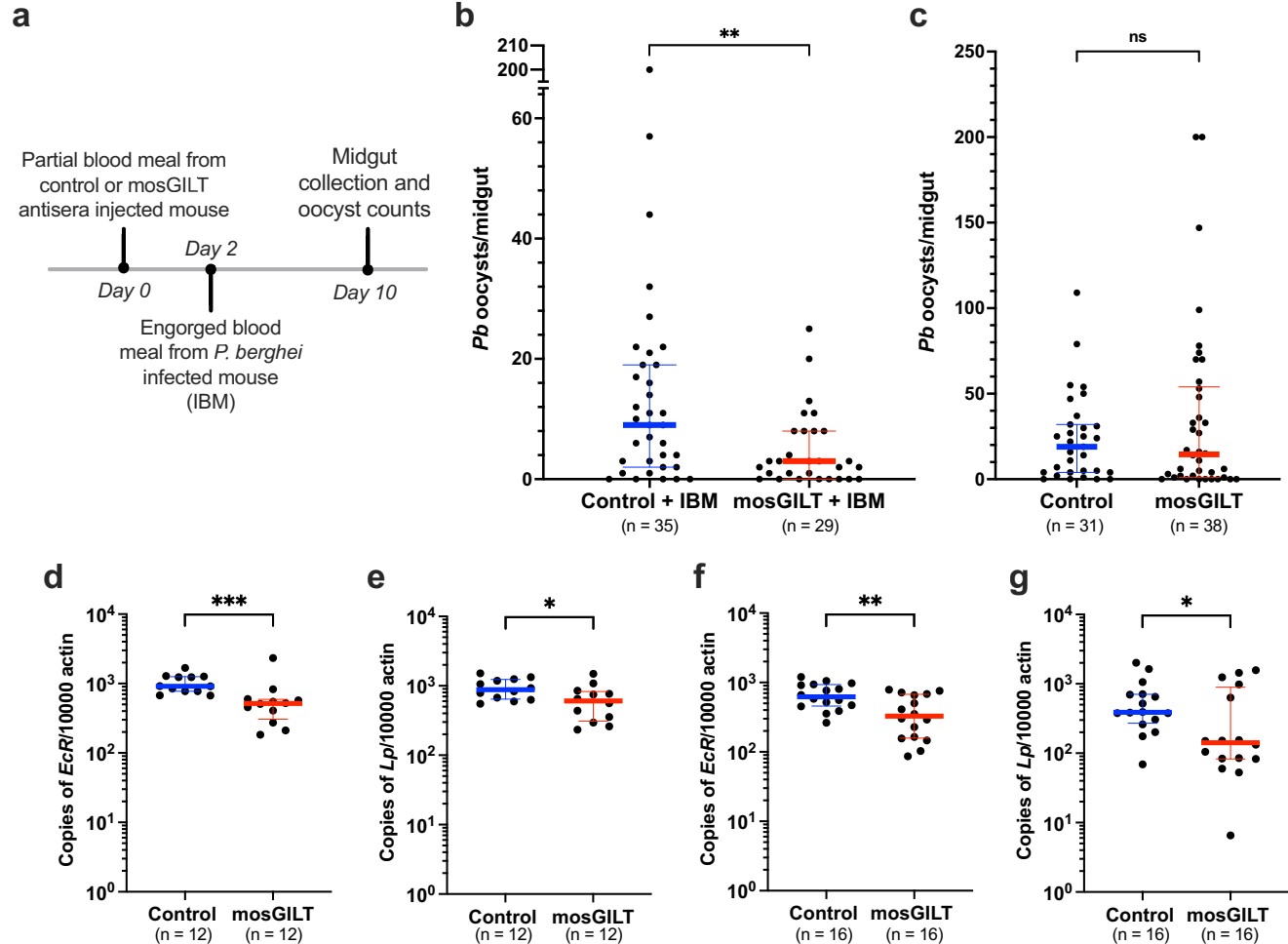

**Fig. 7 | MosGILT antibodies function in the midgut and decrease the expression of the *ecdysone receptor* (*EcR*) and *lipophorin* (*Lp*). a** Summary of the experimental scheme of a partial immunized blood meal followed by infected blood meal (IBM). **b** Number of *P. berghei* oocysts from mosquitoes that took a partial blood meal on a GST (Control + IBM; $n = 35$), or mosGILT (mosGILT + IBM, $n = 29$) antisera immunized mouse followed by an IBM 48 h later ($p = 0.0060$). **c** Number of *P. berghei* oocysts from mosquitoes that engorged on an IBM 48 h after intrathoracic injection with 1.4 μg of purified mosGILT ($n = 38$) or control ($n = 31$) antibodies ($p = 0.9688$). Oocysts were counted eight days post-IBM. The *EcR* (**d**) and *Lp* (**e**) expression level in midguts 48 h after engorgement on animals immunized with

mosGILT ($n = 12$) or control ($n = 12$) antibodies, as determined by RT-qPCR (d, $p = 0.0007$; e, $p = 0.0449$). The expression levels of *EcR* (**f**) and *Lp* (**g**) 48 h after a blood meal on a mouse before (control; $n = 16$) or after retro-orbital injection with mosGILT antibodies ($n = 16$), as determined by RT-qPCR (f, $p = 0.0058$; g, $p = 0.0387$). Each dot represents an individual mosquito midgut, and the blue and red horizontal lines indicate the medians for the control and intervention groups, respectively. The thin lines indicate upper and lower quartiles. At least wo experimental replicates were performed for each experiment. Two-Tailed Mann-Whitney was used to determine significance (ns: not significant, *$p \leq 0.05$, **$p \leq 0.01$, ***$p \leq 0.001$). Source data are provided as a Source Data file.

supplemented with 0.5% penicillin/streptomycin which was subsequently changed every three days. Seven days post-infection mosquito midguts were collected, and the number of oocysts was determined with an EVOS fluorescence microscope. This experiment was replicated more than three times.

### Pfs25 monoclonal antibody

Monoclonal Anti-Plasmodium falciparum 25-kDa Gamete Surface Protein (Pfs25), Clone 4B7 (produced in vitro), MRA-28, contributed by Louis H. Miller and Allan Saul, was obtained through BEI Resources, NIAID, NIH.

### Passive immunization studies

For the passive immunization studies, two mice with comparable parasitemia were selected and administered 400 μl mosGILT or glutathione-S-transferase (GST) antisera via intraperitoneal injection. 24 h later, mosquitoes deprived of sucrose for 18 h were allowed to feed on the mouse. Engorged mosquitoes were sorted and

immediately transferred to a chamber set to 20 °C and maintained as described above. Seven days post-infection, mosquito midguts were collected, and the number of oocysts was determined with an EVOS fluorescence microscope. Three experimental replicates were performed.

To minimize any potential variability in parasitemia, additional experiments were performed where one mouse was selected to feed the control and intervention groups. Mosquitoes deprived of sucrose for 24 h were fed on an anesthetized mouse with ~2% parasitemia. Immediately afterward, the infected mouse was immunized with 100 μl of purified mosGILT antibodies (0.2 mg/ml) or mosGILT monoclonal antibody (0.05 mg/ml) via retro-orbital injection. One hour after immunization, mosquitoes were allowed to feed on the animal. Engorged mosquitoes were immediately transferred to a chamber set to 20 °C and maintained as previously described. Seven days post-infection, midguts were collected, and the number of oocysts was determined with an EVOS fluorescence microscope. Two experimental replicates were performed.

**Partial feeding studies.** Two cages of *A. gambiae* (~200 mosquitoes/cage) had 10% sucrose removed. After 18 h of starvation, a mosGILT antisera-immunized mouse was placed in one cage, and control (GST) antisera-immunized mouse in the other. Passive immunization was completed as described previously. During the feeding period, mice were moved to different locations on the cage every 1–2 min to disrupt feeding and increase the number of partially fed individuals. Following feeding, partially fed mosquitoes were moved to small paper cups covered by a mesh net. Cups were held at 27 °C/80% humidity without sucrose. 48 h later, both groups were allowed to feed on a singular *P. berghei*-infected mouse five or six days post-infection. Murine infection was completed as described previously. Following the ~1 h feeding period, engorged mosquitoes were placed back into paper cups and then moved to 21 °C/80% humidity with sucrose. Midgut collection and oocyst counts were completed eight days post-infected blood meal. The number of oocysts was determined with an EVOS fluorescence microscope. A variation on this experimental scheme was completed where the first blood meal was from a singular *P. berghei*-infected mouse, and the second was from the passively immunized mice. Furthermore, following the first blood meal, mosquitoes were transferred to 21 °C. Two experimental replicates were performed.

**Intrathoracic injection.** *A. gambiae* 3–5 days post-emergence were anesthetized on ice. Mosquitoes were injected in the thorax with either 0.14 µl OVA or mosGILT purified polyclonal antibody (10 µg/µl) with a Drummond Scientific II nanoinjector. Following injection, mosquitoes were placed into small paper cups covered by mesh, a 10% sucrose-soaked cotton pad, a moist paper towel, and plastic wrap. The sucrose pad was removed 36 h post-injection to starve mosquitoes. 48 h post-injection, both groups were allowed to feed on a single infected mouse 5 or 6 days after *P. berghei* infection. Murine infection was completed as described previously. Midgut collection and oocyst counts were completed eight days post-infected blood meal. Three experimental replicates were performed.

**Oocyst counts.** *P. falciparum* midguts were dissected in PBS seven days after infection and stained with 0.2% mercurochrome (M7011; Sigma-Aldrich) in PBS. As previously described, the number of oocysts in the midguts was determined using a phase-contrast microscope (Leica)[64]. *P. berghei* midguts were dissected in PBS seven or eight days post-infected blood meal and examined with an EVOS fluorescence microscope. The number of oocysts in the midguts was determined using GFP as a marker.

**Ookinete counts.** Mosquitoes and mice were treated the same as in the antibody blocking assays of *P. berghei* infection. However, 18 h post blood meal, midguts were dissected and placed into a well of a 96-well plate with 30 µl of PBS. Midguts were homogenized with a pipette. 10 µl of the final homogenate were dispersed thinly within the 15 mm diameter ring of a fluorescent antibody microscope slide (Thermo Fisher Scientific) to form a blood meal smear. Slides were fixed and stained with Giemsa. Each ring was divided into eight sections. Three sections of each smear were randomly selected for ookinete counting using a phase-contrast microscope. The mean per section for each smear was calculated and multiplied by eight to estimate the total number of ookinetes in the smear. This value was multiplied by three to estimate the total number of ookinetes both in the blood bolus and traversing the midgut epithelium for each mosquito. Two experimental replicates were performed.

**mosGILT S2 cells protein expression**
Protein was expressed as previously described[66]. Briefly, S2 cells expressing recombinant mosquito gamma-interferon-inducible lysosomal thiol reductase (rmosGILT), were cultured in serum-free Schneider's medium supplemented with 1% penicillin/streptomycin. The cell cultures were maintained in a spinner flask at room temperature following a 16- to 20-hour adjustment period to serum-free conditions. Induction of protein expression was achieved by adding 500 µM copper sulfate to the culture, and four to five days post-induction, the cell culture was centrifuged to collect the supernatant containing soluble rmosGILT. The supernatant underwent filtration with a 0.2 µm bottle top filter, and Tween-20 (0.05%) and beta-mercaptoethanol (5 mM) were added. Purification of rGILT was carried out using a Ni-NTA agarose column. Elution was performed in a buffer composed of 50 mM NaH2PO4, 500 mM NaCl, and 250 mM imidazole. The purified rGILT was concentrated using a 15 ml 3 kD Amicon filter, followed by three washes with PBS (Fig. S6a). Protein concentrations were quantified using a BCA assay with bovine serum albumin (BSA) standard curve.

**mosGILT bacterial protein expression**
Protein was expressed as previously described[66]. Briefly, mosGILT without its native signal peptide was inserted into the pGEX-6-P2 vector (GE Healthcare) using BamHI and NotI restriction sites and transformed into BL21 chemically competent cells (Thermo Fisher Scientific). After induction with 0.1 mM IPTG at 30 °C for 4 h, cells were lysed and the GST-mosGILT fusion protein was isolated from the lysate using glutathione sepharose 4B (GE Healthcare). The purified protein was then concentrated and desalted using a 3 kD Amicon filter (EMD Millipore) and washed three times with PBS.

**mosGILT antisera and monoclonal antibody generation**
For mosGILT antisera generation GST-mosGILT fusion protein was emulsified in complete Freund's adjuvant (CFA) and injected subcutaneously into a rabbit (200 µg per animal per injection). The rabbit was boosted twice at 2-week intervals with GST-mosGILT fusion protein and incomplete Freund's adjuvant (Thermo Fisher Scientific). Serum was collected two weeks after the last boost. Cocalico Biologicals performed immunizations and serum collection. Antibody responses were determined using enzyme-linked immunosorbent assay and immunoblot. Polyclonal rabbit immunoglobulin G against GST-mosGILT fusion protein was purified using 1 ml of NAb Protein A/G spin columns (Thermo Fisher Scientific). Monoclonal antibodies specific to mosGILT were generated by GenScript (Piscataway, NJ) using purified rGILT, as previously described[66]. 4C9H9 was maintained in DMEM with 10% FBS and 1% penicillin/streptomycin at 37 °C and 5% CO₂. For monoclonal antibody purification, cells were pooled, spun down at 500 x g for 5 min, and transferred into serum-free hybridoma medium (Thermo Fisher Scientific) for seven days at 37 °C and 5% CO₂. The cells were removed from the supernatant by centrifugation at 500× g for 5 min. The supernatant was diluted 1:1 with Protein A/G binding buffer (Thermo Fisher Scientific). 4C9H9 monoclonal antibody was purified using protein A/G agarose (Thermo Fisher Scientific) and concentrated in PBS with a 50 kD Amicon filter. The relative reactivity of these materials to recombinant mosGILT can be seen in Fig. S6b.

**Oocyst cross-sectional area**
Oocyst cross-sectional area was calculated from mosquitoes that took a *P. falciparum* or *P. berghei* infected blood meal from a membrane feeder with 0.2 mg/ml of mosGILT or control antibodies. Midguts were dissected 7 days post infected blood meal. *P. berghei* oocysts were imaged using an EVOS fluorescence microscope. *P. falciparum*-infected midguts were Mercurochrome stained and imaged using a microscope. Oocyst cross-sectional area was measured using these scaled images in FIJI. *P. falciparum* and *P. berghei* experiments were replicated a total of three times.

**Active immunization**
C57BL/6 mice (6-week-old females, Charles River) were immunized once with either 10 µg of rGILT or OVA emulsified in Complete Freud's adjuvant (CFA; Thermo Fisher Scientific) followed by two boosts of the

respective antigen within Incomplete Freud's adjuvant (IFA; Thermo Fisher Scientific) every 14 days via intradermal injection. Fourteen days after the final boost, 25 μl of serum was collected from each mouse, and ELISA tested the titers to confirm antigen-specific antibodies for each mouse. Mice with strong titers were challenged with *P. berghei*-infected red blood cells by intraperitoneal injection. Mosquitoes were allowed to feed five or six days after *P. berghei* infection. Seven days post-infected blood meal mosquito midguts were collected, and the number of oocysts was determined with an EVOS fluorescence microscope. Three experimental replicates were performed.

### Immunostaining

*P. berghei–infected* mosquito midguts were dissected 5 days post-infected blood meal into 1% paraformaldehyde (PFA) in PBS. A small incision in the anterior of each midgut was made. Midguts were then fixed overnight at 4 °C in 4% PFA. After washing three times with PBS, midguts were blocked and permeabilized for 1 h at room temperature in PBS containing 0.2% BSA, 0.1% Triton X100, and 10% goat serum. Midguts were washed three times with PBS and incubated overnight at 4 °C with OVA or mosGILT purified mouse polyclonal antibody (1:500) in PBS containing 0.2% BSA and 0.1% Triton X100. Midguts were washed three times with PBS and incubated with goat anti-mouse Alexa Fluor 647 secondary antibody (1:1000; A21240; Thermo Fisher Scientific) for 4 h at room temperature. Midguts were washed three times with PBS and incubated in Hoechst 33342 (1:1000) (Thermo Fisher Scientific) for 10 min. Immediately after incubation, the midguts were washed one time with PBS and transferred to slides with Aqua-Poly/Mount (Polyscience, Inc). Slides were imaged using a Leica SP8 STED confocal microscope. Four replicates of at least three midguts from the mosGILT and control antibodies were analyzed. Images were processed using ImageJ (FIJI).

### Ovarian development, egg laying capacity, and hatch rate

A mosGILT antisera-immunized mouse was placed on the top of one cage, and control (GST) antisera-immunized mouse on the other. Passive immunization was completed as described previously. Following feeding, the cages were sorted to remove unfed and partially fed mosquitoes. 48 h following the blood meal, ovaries of mosquitoes from each group were dissected an imaged using a Zeiss microscope. At this 48-hour time mark, mosquitoes from each group were also moved to individual drosophila tubes with ~5.0 mL of water. 5 days following the blood meal, the number of eggs and larvae in each tube was counted. Two experimental replicates were performed.

### Gene expression

The RNeasy mini kit (QUIAGEN, CA) was used to extract RNA from individual midguts, 48 h after mosquitoes fed on mice passively administered mosGILT or GST antibody. All extractions followed the manufacturer's protocols. cDNA was made with an iScript kit (Bio-Rad, CA) from RNA. Real-time quantitative PCR was performed using iTaq SYBR Green Supermix (Bio-Rad, CA) on a CFX96 real-time platform (Bio-Rad). The relative expression of different mosquito genes (Table S1) was normalized to the *A. gambiae actin* mRNA (Table S2). These data were presented as the copy number of the target gene per 10,000 copies of the housekeeping gene, *actin*. The primers used for expression of genes are listed in Tables S1 and S2. Two experimental replicates were performed.

### Hemolymph collection and apolipophorin I (*Lp*) quantification

Mosquitoes were anesthetized on ice and transferred to a glass slide. The proboscis was clipped using dissection scissors, and forceps gently pressed the lateral aspects of the thorax to push out a drop of hemolymph from the cut proboscis[66,68]. Hemolymph droplets from 15 mosquitoes fed on a mosGILT or control (GST) antisera immunized mice were collected into Laemmli sample buffer using a pipette tip.

Proteins in the hemolymph were separated by SDS-PAGE and stained with Coomassie blue. To quantify the percentage of *Lp* reduction seen on the SDS-PAGE gel, the intensity of the *Lp* band was measured using ImageJ (FIJI). The expression of *Lp* was normalized using four protein bands that were consistent in both groups. The control protein bands had a size between 50 and 25 kD. This experiment was repeated twice. Two experimental replicates were performed.

### Western blots

Salivary glands and midguts were collected from *A. gambiae* (4arr strain) in PBS. Then tissues were homogenized and resolved by SDS-PAGE using 4–20% Mini-Protean TGX gels (Bio-Rad) at 150 V for 45 to 60 min and then transferred onto a 0.2-μm nitrocellulose membrane for 60 min at 100 V in a Tris-Glycine transfer buffer with 25% methanol. Blots were blocked in 5% nonfat milk in PBS containing 0.1% Tween-20 for 60 min at room temperature. Primary antibodies were diluted in blocking buffer and incubated with the blots for 1 h at room temperature or 4 °C overnight (Rabbit mosGILT polyclonal antibodies 1:10,000). After this, blots were washed three times in PBS with 0.1% Tween-20 three times for 5 min. Secondary antibodies, goat anti-rabbit IgG (H + L) HRP conjugated (Invitrogen), were diluted in blocking buffer at 1:10,000 and incubated for 1 h at room temperature. Blots were washed in PBS with 0.1% Tween-20 three for 5 min and then imaged with a LI-COR Odyssey Fc imaging system.

### Reporting summary

Further information on research design is available in the Nature Portfolio Reporting Summary linked to this article.

## Data availability

All data supporting the findings of this study are available within the article and its Supplementary Information file. Source data are provided with this paper.

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

## Acknowledgements

This work was partly supported by the Howard Hughes Medical Institute Emerging Pathogens Initiative, NIH grants AI142708 (E.F.) and AI158615 (E.F. & G.D.), and Bloomberg Philanthropies (G.D.). We thank the Johns Hopkins Malaria Research Institute Parasite and Insectary core facilities for *Plasmodium falciparum* gametocytes cultures and *Anopheles gambiae* Keele strain mosquito-rearing. We also thank the Yale Center for Cellular & Molecular Imaging Confocal Facility for usage of their Leica SP8 STED microscope.

## Author contributions

Conceptualization, T.C.G., B.D., Y.-M.C., E.F.; Methodology, B.D., T.C.G., Y.D., and Y.-M.C.; Investigation, B.D., T.C.G., Y.D., and Y.-M.C; Resources, P.C.; Writing – Original Draft, T.C.G., B.D., E.F.; Writing – Review & Editing, B.D., T.C.G., Y.D., P.C., G.D., Y.-M.C., and E.F.; Funding Acquisition, E.F. and G.D.; Supervision, E.F. and G.D.

## Competing interests

The authors declare no competing interests.
