## [Transparent Peer Review file · Nature Communications]

mosGILT antibodies interfere with Plasmodium sporogony in Anopheles gambiae

Corresponding Author: Dr Erol Fikrig

Version 0:

Reviewer comments:

Reviewer #1

(Remarks to the Author)

This paper reports that antibodies against mosGILT reduce plasmodium infection levels in vector mosquitoes. Accordingly, this protein joins the very short list of mosquito-derived transmission blocking vaccine targets. The study is well carried out, results are clearly presented, and data about the affected parasite life stages and of the potential mechanisms behind the TB activity are included.

Perhaps the authors can elucidate on the localisation of mosGILT in the midgut (Fig. 1b) ?

Reviewer #2

(Remarks to the Author)

Summary: authors' report findings that mosGILT specific antibodies interfere with sexual development within the mosquito host of malaria parasites.

Comments:

1. Introduction:

a. suggest revision of section discussing TB31F which acts on gametes in the mosquito midgut and not gametocytes which are within RBCs. In addition, suggest identifying that TB31F targets Pfs48/45 for clarification.

2. Results and discussion:

a. Overall discussion point - important to understand and control for assay variation when oocyst counts are low, see Swihart, Fay and Miura PMID: 31007315. Mean oocyst counts of 1.1 likely bias the interpretation of transmission blocking activity. It is also important to indicate the weak association between TRA and TBA or reduced prevalence.

b. It is difficult to assess the impact of antibody concentrations against mosGILT and biological activity within the various studies. Would it be possible to perform ELISAs on different materials to establish a comparative measure? Also, suggest inclusion of a known transmission blocking candidate such as Pfs25 or Pfs230 in performing P. falciparum studies for TRA comparison. Given the cost of development, an understanding of a rank order of biological activity is valuable to decision making.

c. Suggest inclusion of the expression, and biochemical and biophysical characterization of recombinant mosGILT. The quality of the recombinant protein impacts the specificity and quality of the antibodies generated.

d. The association shown between RNA transcript levels and protein expression in the supplemental information is insufficient to support the authors' position. Additional proteomic studies are required to clearly show a reduction of protein expression impacts the biological outcomes. The discussion draws conclusions based on ungrounded results.

e. During the evaluation of transmission reduction studies, were samples prepared for histological studies? Histological studies seem possible and would provide support for the authors' story.

f. Authors should include a discussion of a comparative analysis between human and mosGILT and the implications of targeting mosGILT as a vaccine candidate. Even though the homology may be considered low (~28% identity based on quick comparison), the generation of cross-reactive antibodies will be a concern.

Reviewer #3

(Remarks to the Author)

The manuscript by Correa Gaviria and colleagues, entitled “mosGILT antibodies interfere with Plasmodium sporogony in Anopheles gambiae” investigates the impact of mosquito interferon- γ -inducible lysosomal thiol reductase (mosGILT) antibodies in the development of Plasmodium within the mosquito midgut. Here are their major findings:

1. The study initially shows that mosGILT mRNA levels are transiently elevated after a blood meal and the protein is located not only in the salivary glands (previously published), but also in the midgut a key site for the Plasmodium sporogonic cycle.
2. The authors demonstrate the impact of mosGILT antibodies on reducing the prevalence of Plasmodium infection, which is the most significant finding of this work. Specifically, the study shows that antibodies against mosGILT significantly reduce the median total number of *P. falciparum* and *P. berghei* oocysts in the mosquito midgut. In addition, mosGILT antibodies decrease the infection prevalence of both parasite species in the mosquito, highlighting their potential as a transmission-blocking strategy. These findings were validated using both membrane-feeding and in vivo murine infection models that employed both active and passive immunization interventions.
3. The authors' findings also suggest that the midgut, rather than the hemolymph, and the oocyst stage is the primary site of action for mosGILT antibodies.
4. Finally, this study found that administration of mosGILT antibodies decrease the expression of the ecdysone receptor (EcR) and lipophorin (Lp) at the mRNA level. EcR and Lp function in both immunity against and nutrition of malaria parasites (published by others).

This work is original in its approach to using mosGILT antibodies as a transmission-blocking strategy. While previous studies for the same laboratories have explored the role of mosGILT in Plasmodium infection and mosquito biology using mosaic gene knockout strategies, the application of antibodies to target this protein for malaria control is a novel contribution. Overall, the research highlights the potential of mosGILT as a target for TBVs, which could complement existing malaria control strategies.

However, the study is limited in novelty and scope, and more importantly, the experimental setup used raises concerns about the validity of the authors' major conclusions. In addition, the manuscript is not rigorous, as authors do not provide the primary datasets that would allow validation of their conclusions, nor does the Materials and Methods section provide sufficient detail to enable readers to replicate the experiments. Lastly, the biological function is derived from very general phenotypic descriptions and RNA seq results, which are helpful to form hypotheses, which however should be tested rigorously before stating mechanisms of action.

Major comments:

-All mosquitoes were fed on a mixture of 10% sucrose and 5% penicillin/streptomycin, which is likely to lead to a serious disruption of the microbiota. Several studies (including studies published by authors of this study) have shown that dysbiosis alters Plasmodium infection as well as immune responses in the midgut. Without showing controls that the antibiotic treatment does not affect the infection outcome in the control or experimental treatments, the major finding of this manuscript is questionable.

-There are several primary datasets missing in the manuscript:

1. The authors must include the original infection data in table format (replicate number, treatment group, number of oocysts per midgut). Otherwise, the reader is unable to evaluate the validity of the authors' conclusions.
2. While the authors state that they performed RT-qPCR data for 33 genes, the manuscript only shows the data for two genes and states that there was no difference found between experimental and control mosquitoes for the remaining 31 genes. The authors must include the original RT-qPCR data for all genes in supplement. Otherwise, the reader is unable to evaluate the validity of the authors' conclusions.

-The conclusion that the anti-mosGILT antibodies act on the midgut (in the abstract more specifically referred to as midgut lumen) and impact the oocyst stage seem contradictory and also are not supported by the data provided. The authors base their conclusion on three findings, respectively: 1. Administration of anti-mosGILT antibodies 48h after infection reduces overall “parasite load” (see point 4 below). 2. Injection of anti-mosGILT antibodies 48h prior infection does not reduce overall “parasite load”. 3. similar ookinete counts between treated and control mosquitoes at 18h post infection. However, all three findings are up for interpretation (none of which are not discussed by the authors) and at best provide indirect evidence for their conclusions, for the following reasons:

- Partial feeding leads to highly variable parasite numbers, as the ingested dose is difficult if not impossible to control. Without bloodmeal size analysis, how can the authors be sure that any observed differences are not due to difference in ingested parasite dose?
- the authors ignore alternative explanations for the lack of impact of injected antibodies on parasite development, including elevated immune parameters in the control group due to priming by the injection, and potentially lower stability of the injected anti-mosGILT antibodies in the hemolymph.
- The counts of ookinetes, if as described in the Materials and Methods section, does provide total ookinete count, not the count in the midgut lumen, as the authors did not separate blood bolus from the midgut epithelium. The experimental temperature of 20C, required for Plasmodium berghei development during the early stages of the sporogonic cycle within the mosquito, significantly delays blood digestion and excretion of the blood bolus. Thus, sampling midgut ookinetes is insufficient to differentiate between ookinetes that are stuck in the blood bolus and those that are migrating across the midgut epithelium.

-A major component of any transmission-blocking strategy is that the strategy should have little to no impact on mosquito

fitness. The authors determine that the egg laying capacity and larval hatching is unaffected by passive immunization against mosGILT (Figure S2: In panel b). However, the reader cannot evaluate the authors' conclusions, due to the lack of any description in the Materials and Methods section (were these experiments performed at 20C, which is an unrealistic temperature to evaluate fitness parameters for *Anopheles gambiae*). It appears that around 50% of the control females produced non-viable eggs, which seems unusual, and perhaps suggest that these experiments were performed at a low temperature? The authors also describe that "[...] the ovaries [...] were not altered." However, this statement is not supported by data.

Other comments:

-The parasite load data are confounded by prevalence data. The authors should separate infection intensity (number of oocysts per infected mosquito) and prevalence.

-The results section contains details on methods, which are largely repeated in the Materials and Methods section and therefore should be removed from the results section and instead include the rationale for the each of the experiments that were performed.

-The discussion section fails to discuss the results. It is mainly a summary of the results, and does not specifically address Pf and Pb infections. The connection between Lp and EcR downregulation and impact of mosGILT antibodies is unclear, and a more specific approach to identify the effects of mosGILT on the parasite is needed first to then link it to the action of mosGILT antibodies.

-The Materials and Methods section is very succinct, and more detailed descriptions are required, especially in the generation of recombinant proteins and antibodies, gene expression measured by qPCR, and the conditions used in egg laying and hatching experiments.

-The authors refer to Lp throughout the manuscript, and do not distinguish between its subunits of Apolipoprotein I and II. The SDS page bands the authors refer to as Lp in Fig. S3 seem to be apoLp-I (although the size of the marker bands are not defined). Which one was assessed by RT-qPCR?

- Fig.S3 seems to indicate that Vg (according to Rono et al 2010, the band closest to what is presumably apoLp-I) is much reduced in the treatment group? If this is the case, how do the authors explain equal egg laying in treatment and control groups?

Does mosGILT bind to the parasite surface? Previous work by the authors has shown that mosGILT binds to sporozoites in the salivary glands, suggesting it might also bind to other parasite stages. However, the absence of an effect of mosGILT antibodies on ookinete numbers in the lumen could be due to mosGILT not binding to parasites at this stage. Providing direct evidence that mosGILT does not bind to the parasite surface in ookinetes would help clarify why it does not impact luminal stages.

Minor comments:

-In the abstract, the authors state that "Malaria transmission begins when an *Anopheles* mosquito feeds on a *Plasmodium*-infected vertebrate." (Ins 24-25). This is factually incorrect, as non-human vertebrate malaria transmission relies on mosquito species in other genera. Please rephrase.

- Ln 55: Replace "this" for "these" in "This life stages are of interest..".

-Ln 96: In the phrase "This study provides a crucial link between," is "this" referring to the study mentioned in the previous sentence or the current study? Since both studies establish a crucial link between *Plasmodium* sporogony in *A. gambiae* and the mosGILT protein, please rephrase to clearly indicate which study is being referred to.

- In passive immunization studies, including one-mouse studies, how many biological replicates were performed? Please clarify this in the methods section.

- "Intervention" and "experimental" groups are used interchangeably. To maintain consistency, please use either "intervention" or "experimental" throughout the text.

-Figure S1: How many mosquitoes per treatment were evaluated for the ookinete counts?

-Figure S1: There is a typo in the y-axis label for "Pb ookinetes."

-Following up on Fig. 6, it is unclear whether each dot represents the qPCR result from an individual whole mosquito or an individual midgut. The figure legend states "individual mosquito," but the methods section only describes the protocol for midguts (Fig. 1c) and lists the genes analyzed in Fig. 6. Please clarify this in both the figure legend and the methods section.

- In the Lp protein expression experiments, how many replicates were performed?

- Ln 563-564: The word "interferon" is duplicated.

Reviewer #4

(Remarks to the Author)

Version 1:

Reviewer comments:

Reviewer #1

(Remarks to the Author)

Reviewer #2

(Remarks to the Author)

The authors' responses are acceptable. This reviewer agrees that understanding mosquito/parasite biology is important by itself. If the idea moving forward is develop a transmission blocking vaccine then it is important to understand the comparative TRA. The inclusion of the supplemental figure showing the comparative Pfs25 TRA is helpful. It should be understood that additional work will be required for mosGILT vaccine development but this provides a baseline.

Reviewer #3

(Remarks to the Author)

The authors have addressed the majority of our previous concerns.

One of the previous comments remains unsolved, namely the question of whether prevalence (the percent of infected mosquitoes) and/or parasite infection intensity (the number of parasites within a infected mosquito, also referred to as parasite load) is reduced, as the authors claim throughout the manuscript.

The supplementary Datafile 3 shows that prevalence is reduced, which is the goal of any intervention strategy as a very small number of sporozoites is required to establish human infection.

However, in the experimental *Plasmodium falciparum* infections, parasite numbers per infected mosquito (those that have one or more parasites) are not reduced (Figure 2A, B). Thus part of the title of Fig. 2, namely "[...] decrease *P. falciparum* oocyst numbers[...]" is misleading, and so are statements throughout the manuscript that claim that MosGilt significantly reduces both oocyst infection intensity and prevalence.

The text should be modified to clarify the results, namely that prevalence is reduced.

Reviewer #4

(Remarks to the Author)

We thank the editor and reviewers for their suggestions and feedback on how to improve the manuscript. Please see our responses to each comment below.

Reviewer #1 (Remarks to the Author):

This paper reports that antibodies against mosGILT reduce plasmodium infection levels in vector mosquitoes. Accordingly, this protein joins the very short list of mosquito-derived transmission blocking vaccine targets. The study is well carried out, results are clearly presented, and data about the affected parasite life stages and of the potential mechanisms behind the TB activity are included.

Perhaps the authors can elucidate on the localization of mosGILT in the midgut (Fig. 1b)?

We agree and have performed localization studies. During confocal microscopy, the anterior and posterior regions of the midgut were imaged to determine the distribution of protein expression. Based on our existing immunostaining data, mosGILT can be detected in the midgut's anterior and posterior regions. We have incorporated this knowledge into the results (Fig. 1b).

Reviewer #2 (Remarks to the Author):

Summary: authors' report findings that mosGILT specific antibodies interfere with sexual development within the mosquito host of malaria parasites.

Comments:

1. Introduction:

a. Suggest revision of section discussing TB31F which acts on gametes in the mosquito midgut and not gametocytes which are within RBCs. In addition, suggest identifying that TB31F targets Pfs48/45 for clarification.

Thank you for the suggested clarification. We have revised this section of the introduction to make it clear that the TB31F monoclonal antibody targets Pfs48/45 on the surface of *Plasmodium* gametes present in the mosquito midgut.

2. Results and discussion:

a. Overall discussion point - important to understand and control for assay variation when oocyst counts are low, see Swihart, Fay and Miura PMID: 31007315. Mean oocyst counts of 1.1 likely bias the interpretation of transmission blocking activity. It is also important to indicate the weak association between TRA and TBA or reduced prevalence.

We agree that an additional explanation is helpful. Though the control's mean oocyst load is low, analyzing data with such a mean is important as it reflects natural mosquito infection level in the

field, where most infected mosquitoes contain 1-5 oocysts (André et al., JID, 2016) (Gonçalves et al., Nat Commun, 2017). Additionally, the range of mean oocyst intensities present across our various controls (1.1 – 36.5) allows the reader to assess TRA under various conditions.

b. It is difficult to assess the impact of antibody concentrations against mosGILT and biological activity within the various studies. Would it be possible to perform ELISAs on different materials to establish a comparative measure?

We agree and performed ELISAs with mosGILT rabbit and mouse antisera, monoclonal antibody, and purified rabbit IgG and the recombinant mosGILT protein to provide a comparative measure. These data can be found in Fig. S6b.

Also, suggest inclusion of a known transmission blocking candidate such as Pfs25 or Pfs230 in performing *P. falciparum* studies for TRA comparison. Given the cost of development, an understanding of a rank order of biological activity is valuable to decision making.

Thank you for the suggestion. The primary goal of this manuscript is to define the transmission-blocking activity of mosGILT antibodies rather than directly compare their efficacy to antibodies targeting parasite proteins, such as Pfs25. The working mechanisms of transmission-blocking antibodies targeting mosquito proteins differ significantly from those that directly target parasites. Though a TRA comparison between mosGILT and targets like Pfs25 or Pfs230 may be interesting, we believe that such experiments would not yield highly meaningful conclusions at this stage. Future studies should first identify the most effective mosGILT epitopes before assessing their activities against or in combination with existing parasite targets. Readers interested in a rough comparison between mosGILT and established candidates like Pfs25 can consult the existing literature, which we have referenced in the discussion (Mlambo et al., Infect Immun, 2008). However, in response to the reviewer's interest, we have conducted a preliminary *P. falciparum* membrane feeding assay with at least three biological replicates. This included mosGILT rabbit polyclonal IgG at 200 µg/mL alongside a Pfs25 monoclonal antibody (Pfs25 mAb) at 15 µg/mL here. Significant reductions in *P. falciparum* oocyst infection intensity and infection prevalence were observed for both interventions, though, as expected, the Pfs25 mAb provided superior protection to the mosGILT antibodies. The dot plot to the right depicts the oocyst loads in the individual midguts along with

the median number (as shown in the black bar) of *P. falciparum* oocysts per midgut. Infection prevalence of each group is shown in each pie chart below. While we are happy to include this data in the manuscript if the editors and reviewers recommend it, we felt it is more appropriate to keep the discussion general at this stage. Differences in antibody concentrations and preparation methods (pAb vs mAb) levels of antibodies given can influence experimental outcomes for comparison purposes, and we aim to avoid overinterpretation of these results.

c. Suggest inclusion of the expression, and biochemical and biophysical characterization of recombinant mosGILT. The quality of the recombinant protein impacts the specificity and quality of the antibodies generated.

We appreciate this as well. The quality and purity of the recombinant protein (rmosGILT) can be seen in the Coomassie of Fig. S6a. Some biochemical characterization for the recombinant mosGILT protein has been published in the supplemental information of a previous publication (Schleicher et al., Nat Commun, 2018). To summarize some key findings, this work showed recombinant mosGILT, which has an active site of CxxS does not have thiol reducing activity at environmental pH conditions of 4.5 or 7.4. This is in stark contrast to human (active site of CxxC) and mouse GILT C2 (active site mutated from CxxC to CxxS) which had activity at a pH or 4.5. Additionally, mosGILT has an extended C-terminus, when compared to mammalian GILT, that is composed of hydrophobic amino acid residues (Schleicher et al., Nat Commun, 2018).

d. The association shown between RNA transcript levels and protein expression in the supplemental information is insufficient to support the authors' position. Additional proteomic studies are required to clearly show a reduction of protein expression impacts the biological outcomes. The discussion draws conclusions based on ungrounded results.

We agree and have updated the discussion to emphasize that future work should focus on investigating the potential mechanism of how mosGILT antibody elicited decreases in *Lp* and *EcR* levels, ultimately leading to decreased parasite infection and intensity within the mosquito.

e. During the evaluation of transmission reduction studies, were samples prepared for histological studies? Histological studies seem possible and would provide support for the authors' story.

We agree and have included this information. For the standard membrane feeding studies, oocyst structure in the midgut tissue of the control and mosGILT fed groups was evaluated. A 78% and 27% reduction in *P. falciparum* and *P. berghei* oocyst cross-sectional area was observed in mosquitoes that fed on the mosGILT antibody compared to the control. These results are presented in Fig. 3a,b.

f. Authors should include a discussion of a comparative analysis between human and mosGILT and the implications of targeting mosGILT as a vaccine candidate. Even though the homology may be considered low (~28% identity based on quick comparison), the generation of cross-reactive antibodies will be a concern.

We have incorporated this consideration for mosGILT as a vaccine candidate into the discussion.

Reviewer #3 (Remarks to the Author):

The manuscript by Correa Gaviria and colleagues, entitled “mosGILT antibodies interfere with Plasmodium sporogony in Anopheles gambiae” investigates the impact of mosquito interferon- γ -inducible lysosomal thiol reductase (mosGILT) antibodies in the development of Plasmodium within the mosquito midgut. Here are their major findings:

1. The study initially shows that mosGILT mRNA levels are transiently elevated after a blood meal and the protein is located not only in the salivary glands (previously published), but also in the midgut a key site for the Plasmodium sporogonic cycle.
2. The authors demonstrate the impact of mosGILT antibodies on reducing the prevalence of Plasmodium infection, which is the most significant finding of this work. Specifically, the study shows that antibodies against mosGILT significantly reduce the median total number of *P. falciparum* and *P. berghei* oocysts in the mosquito midgut. In addition, mosGILT antibodies decrease the infection prevalence of both parasite species in the mosquito, highlighting their potential as a transmission-blocking strategy. These findings were validated using both membrane-feeding and in vivo murine infection models that employed both active and passive immunization interventions.
3. The authors' findings also suggest that the midgut, rather than the hemolymph, and the oocyst stage is the primary site of action for mosGILT antibodies.
4. Finally, this study found that administration of mosGILT antibodies decrease the expression of the ecdysone receptor (EcR) and lipophorin (Lp) at the mRNA level. EcR and Lp function in both immunity against and nutrition of malaria parasites (published by others).

This work is original in its approach to using mosGILT antibodies as a transmission-blocking strategy. While previous studies for the same laboratories have explored the role of mosGILT in Plasmodium infection and mosquito biology using mosaic gene knockout strategies, the application of antibodies to target this protein for malaria control is a novel contribution. Overall, the research highlights the potential of mosGILT as a target for TBVs, which could complement existing malaria control strategies. However, the study is limited in novelty and scope, and more importantly, the experimental setup used raises concerns about the validity of the authors' major conclusions. In addition, the manuscript is not rigorous, as authors do not provide the primary datasets that would allow validation of their conclusions, nor does the Materials and Methods section provide sufficient detail to enable readers to replicate the experiments. Lastly, the biological function is derived from very general phenotypic descriptions and RNA seq results, which are helpful to form hypotheses, which however should be tested rigorously before stating mechanisms of action.

Major comments:

-All mosquitoes were fed on a mixture of 10% sucrose and 5% penicillin/streptomycin, which is likely to lead to a serious disruption of the microbiota. Several studies

(including studies published by authors of this study) have shown that dysbiosis alters Plasmodium infection as well as immune responses in the midgut. Without showing controls that the antibiotic treatment does not affect the infection outcome in the control or experimental treatments, the major funding of this manuscript is questionable.

We agree that there is substantial literature supporting the role of dysbiosis in altering *Plasmodium* infection as well as immune responses in the mosquito midgut. For *P. falciparum* membrane feeding experiments, the utilized 10% sucrose solution was not supplemented with penicillin/streptomycin, and therefore, there was no concern of microbiota disruption and derived mosquito immune responses in the mosquito midguts for these experiments. However, for *P. berghei* membrane feeding and *in vivo* experiments, penicillin/streptomycin was specifically used due to its ability to favor oocyst development within the mosquito midgut. As noted by Frischknecht et al. (2006), without the use of an antibiotic, it becomes very challenging to achieve sufficient infection rates in mosquitoes to execute an experimental plan effectively (Frischknecht et al., Malar J, 2006). This is one limitation of working with the rodent malaria parasite to study dynamics *in vivo*. We have modified the section on Materials and Methods to clearly describe the details.

-There are several primary datasets missing in the manuscript:

1. The authors must include the original infection data in table format (replicate number, treatment group, number of oocysts per midgut). Otherwise, the reader is unable to evaluate the validity of the authors' conclusions.

Please see "Source_Data." included in our revised manuscript.

2. While the authors state that they performed RT-qPCR data for 33 genes, the manuscript only shows the data for two genes and states that there was no difference found between experimental and control mosquitoes for the remaining 31 genes. The authors must include the original RT-qPCR data for all genes in supplement. Otherwise, the reader is unable to evaluate the validity of the authors' conclusions.

Please see "Source_Data2." included in our revised manuscript and we have included this in the supplemental figure S4.

- Partial feeding leads to highly variable parasite numbers, as the ingested dose is difficult if not impossible to control. Without bloodmeal size analysis, how can the authors be sure that any observed differences are not due to difference in ingested parasite dose?

As pointed out, partial feeding can lead to high variability in parasite numbers due to the large range of blood volumes and ingested parasite numbers. To confirm that the observed differences in our experiments are not due to these factors, blood meal size and ingested parasite dose between partially fed and engorged mosquitoes were compared. Mosquitoes were allowed to feed on a *P. berghei*-infected mouse. Immediately following feeding, mosquitoes were sorted as partially fed or engorged. Midguts were carefully dissected as to not expel midgut contents. qRT-PCR was then used to assess the level of mouse *Actin* and *Pb 18S rRNA* in each midgut. Mouse *Actin* and *Pb 18S rRNA* were used as a proxy for blood meal size and ingested parasite dose, respectively. As can be seen by the preliminary data below, which is the sum of two independent experimental replicates, there is no evidence to suggest that mosquitoes assigned a partially fed status in our two experiments have higher variability in blood meal size or ingested parasite dose compared to mosquitoes assigned an engorged status. This finding is likely due to strict sorting criteria when assigning a mosquito to the partially fed group. Visual representations of the criteria used during sorting has been added to Fig. 5. Furthermore, due to the random assignment of mosquitoes to the control (control antibody) or intervention groups (mosGILT antibody), any existing variability, partial or engorged, is assumed to be even in both groups.

- The authors ignore alternative explanations for the lack of impact of injected antibodies on parasite development, including elevated immune parameters in the control group due to priming by the injection, and potentially lower stability of the injected anti-mosGILT antibodies in the hemolymph.

These potential alternative explanations have been incorporated into the discussion.

- The counts of ookinetes, if as described in the Materials and Methods section, do provide the total ookinete count, not the count in the midgut lumen, as the authors did not separate blood bolus from the midgut epithelium. The experimental temperature of 20C, required for *Plasmodium berghei* development during the early stages of the sporogonic cycle within the mosquito, significantly delays blood digestion and excretion of the blood bolus. Thus, sampling midgut ookinetes is insufficient to differentiate between ookinetes that are stuck in the blood bolus and those that are migrating across the midgut epithelium.

We wanted to determine if total ookinete abundance was being impacted directly by the mosGILT antibody. To do this, midguts were sampled 18 hours post-infected blood meal to determine the total number of ookinetes in each mosquito (ookinetes in the blood bolus and those traversing the midgut epithelium) – this number was found to be unaffected. Furthermore, partial feeding studies (infected blood meal followed by mosGILT immunized blood meal) suggest that the mosGILT antibody is unlikely to work on ookinete stages. The Results, Discussion, Materials and Methods, and Supplemental Materials sections have been updated to clarify this point.

-A major component of any transmission-blocking strategy is that the strategy should have little to no impact on mosquito fitness. The authors determine that the egg laying capacity and larval hatching is unaffected by passive immunization against mosGILT (Figure S2: In panel b). However, the reader cannot evaluate the authors' conclusions, due to the lack of any description in the Materials and Methods section (were these experiments performed at 20C, which is an unrealistic temperature to evaluate fitness parameters for *Anopheles gambiae*). It appears that around 50% of the control females produced non-viable eggs, which seems unusual, and perhaps suggest that these experiments were performed at a low temperature? The authors also describe that "[...] the ovaries [...] were not altered." However, this statement is not supported by data.

These experiments were repeated at 27 °C to more accurately assess the potential impact of the mosGILT antibodies on egg lay and larval hatch rate. Additionally, images of the ovaries were captured 2 days post-blood meal. Results can be found in Fig. S2 and Fig. S3. Based on all our findings, the mosGILT antibody does not negatively impact female mosquito reproductive fitness. Additionally, a description of this methodology has been added to the Materials and Methods section.

Other comments:

-The parasite load data are confounded by prevalence data. The authors should separate infection intensity (number of oocysts per infected mosquito) and prevalence.

Suppose the parasite load is presented in the way described above. In that case, the one challenge arises is when readers may want to compare this mosGILT transmission-blocking data to related published data – one such comparison could be with Pfs25, as Reviewer #2 suggested. Many publications on the TBA/TRA of Pfs25 antibodies include zeros when determining parasite load (Duffy et al., *Infect Immun*, 1997) (Stowers et al., *Infect Immun*, 2000) (de Graaf et al., *Front Immunol*, 2021). Therefore, we prefer to present the data as shown. However, to address this concern, a detailed statistical table with p-values—both with and without zeros— has been added to the supplementary materials and can be found in “Data_File_S3.”.

-The results section contains details on methods, which are largely repeated in the Materials and Methods section and therefore should be removed from the results section and instead include the rationale for the each of the experiments that were performed.

Details of the methodology have been removed from the Results section.

-The discussion section fails to discuss the results. It is mainly a summary of the results, and does not specifically address Pf and Pb infections. The connection between Lp and EcR downregulation and impact of mosGILT antibodies is unclear, and a more specific approach to identify the effects of mosGILT on the parasite is needed first to then link it to the action of mosGILT antibodies.

The results and discussion have been modified to hopefully clarify the potential connection between reduced EcR and Lp protein levels and the strong reduction in parasite intensity and infection. To also clarify the point here, previous studies have shown that when *EcR* was co-silenced with *Lp* via RNAi, resulting in reduced protein levels in the midgut, there was a significant decrease in both oocyst number and size - a reverse phenotype from when EcR was silenced alone. This finding, as pointed out by the authors, suggests that lipid trafficking by Lp is crucial for oocyst development (Werling et al., Cell, 2019). This finding aligns with the transcriptional, protein, and oocyst phenotypes we observed with the mosGILT antibodies. Future work needs to be done to understand better the mechanism(s) driving these phenotypes.

-The Materials and Methods section is very succinct, and more detailed descriptions are required, especially in the generation of recombinant proteins and antibodies, gene expression measured by qPCR, and the conditions used in egg laying and hatching experiments.

Additional details have been added to the methods sections describing the generation of recombinant protein and antibodies and the gene expression measured by qRT-PCR. An additional section to the Materials and Methods was added to discuss the conditions used in egg laying and hatching experiments.

-The authors refer to Lp throughout the manuscript, and do not distinguish between its subunits of Apolipoprotein I and II. The SDS page bands the authors refer to as Lp in Fig. S3 seem to be apoLp-I (although the size of the marker bands are not defined). Which one was assessed by RT-qPCR?

Yes, we assessed the ApoLp-I band on the SDS page. The results and discussion clarify this distinction. For RT-qPCR analysis, ApoLpII/I (AGAP001826) was assessed (Supplementary Table 2).

- Fig.S3 seems to indicate that Vg (according to Rono et al 2010, the band closest to what is presumably apoLp-I) is much reduced in the treatment group? If this is the case, how do the authors explain equal egg laying in treatment and control groups?

We agree; based on the SDS Page bands, Vg does appear to be reduced in the mosGILT group compared to the control at Days 3 & 4 post antibody ingestion. Despite this finding, no impact of the antibody on ovaries, egg laying, or hatch was observed. This phenotype could be explained by the difference between peak *Vg* expression and observed impacts to *Vg* protein levels by the antibody. As discussed by Stryapunina et al. (2024), *Vg* peaks 24 hours post blood meal and provides amino acids to oocytes. Since no *Vg* reduction is observed by the SDS Page until 72 hours post mosGILT antibody blood meal, it is possible that *Vg* protein levels were sufficient during this critical period for oocyte development and, therefore, the potential reduction at these later time points does not impact egg development (Stryapunina et al., PloS Genet, 2024). That being said, additional inquiry is needed to better understand this finding and provide a robust explanation.

Does mosGILT bind to the parasite surface? Previous work by the authors has shown

that mosGILT binds to sporozoites in the salivary glands, suggesting it might also bind to other parasite stages. However, the absence of an effect of mosGILT antibodies on ookinete numbers in the lumen could be due to mosGILT not binding to parasites at this stage. Providing direct evidence that mosGILT does not bind to the parasite surface in ookinetes would help clarify why it does not impact luminal stages.

This is a very interesting question that we aim to address in future experiments.

Minor comments:

-In the abstract, the authors state that “Malaria transmission begins when an Anopheles mosquito feeds on a Plasmodium-infected vertebrate.” (lns 24-25). This is factually incorrect, as non-human vertebrate malaria transmission relies on mosquito species in other genera. Please rephrase.

Agree and changed accordingly.

- Ln 55: Replace “this” for “these” in “This life stages are of interest..”.

Changed to “these”

-Ln 96: In the phrase “This study provides a crucial link between,” is “this” referring to the study mentioned in the previous sentence or the current study? Since both studies establish a crucial link between Plasmodium sporogony in *A. gambiae* and the mosGILT protein, please rephrase to clearly indicate which study is being referred to.

The sentence has been rephrased to indicate that this thought refers to the study discussed in the prior sentence.

- In passive immunization studies, including one-mouse studies, how many biological replicates were performed? Please clarify this in the methods section.

Three replicates were performed under antisera passive immunization, two replicates were performed under the one-mouse model with purified rabbit mosGILT polyclonal antibodies, and two replicates were performed under the one-mouse model with the monoclonal antibody. This information has been clarified in the Materials and Methods section.

-“Intervention” and “experimental” groups are used interchangeably. To maintain consistency, please use either "intervention" or "experimental" throughout the text.

Agree. Now consistent throughout the manuscript.

-Figure S1: How many mosquitoes per treatment were evaluated for the ookinete counts?

38 mosquitoes per treatment were evaluated for the ookinete counts. The number of samples per treatment have been added to all figures.

-Figure S1: There is a typo in the y-axis label for “Pb ookinetes.”

Changed to “*Pb* ookinetes”

-Following up on Fig. 6, it is unclear whether each dot represents the qPCR result from an individual whole mosquito or an individual midgut. The figure legend states “individual mosquito,” but the methods section only describes the protocol for midguts (Fig. 1c) and lists the genes analyzed in Fig. 6. Please clarify this in both the figure legend and the methods section.

Language has been consistent across the figure legend and methods section to make it clear that each dot represents the qPCR result from an individual midgut.

- In the Lp protein expression experiments, how many replicates were performed?

Two experiment replicates were completed.

- Ln 563-564: The word “interferon” is duplicated.

“Interferon” has been removed.

Reviewer #4 (Remarks to the Author):

References

- 1) André Lin Ouédraogo, Bronner P. Gonçalves, Awa Gnémé, Edward A. Wenger, Moussa W. Guelbeogo, Amathe Ouédraogo, Jaline Gerardin, Caitlin A. Bever, Hil Lyons, Xavier Pitroipa, Jan Peter Verhave, Philip A. Eckhoff, Chris Drakeley, Robert Sauerwein, Adrian J. F. Luty, Bocar Kouyaté, Teun Bousema, Dynamics of the Human Infectious Reservoir for Malaria Determined by Mosquito Feeding Assays and Ultrasensitive Malaria Diagnosis in Burkina Faso, *The Journal of Infectious Diseases*, Volume 213, Issue 1, 1 January 2016, Pages 90–99, <https://doi.org/10.1093/infdis/jiv370>
- 2) Gonçalves, B.P., Kapulu, M.C., Sawa, P. et al. Examining the human infectious reservoir for *Plasmodium falciparum* malaria in areas of differing transmission intensity. *Nat Commun* 8, 1133 (2017). <https://doi.org/10.1038/s41467-017-01270-4>
- 3) Mlambo G, Maciel J, Kumar N2008. Murine Model for Assessment of *Plasmodium falciparum* Transmission-Blocking Vaccine Using Transgenic *Plasmodium berghei* Parasites Expressing the Target Antigen Pfs25. *Infect Immun* 76: <https://doi.org/10.1128/iai.01409-07>
- 4) Schleicher, T.R., Yang, J., Freudzon, M. et al. A mosquito salivary gland protein partially inhibits *Plasmodium* sporozoite cell traversal and transmission. *Nat Commun* 9, 2908 (2018). <https://doi.org/10.1038/s41467-018-05374-3>
- 5) Frischknecht, F., Martin, B., Thiery, I. et al. Using green fluorescent malaria parasites to screen for permissive vector mosquitoes. *Malar J* 5, 23 (2006). <https://doi.org/10.1186/1475-2875-5-23>
- 6) Duffy PE, Kaslow DC. 1997. A novel malaria protein, Pfs28, and Pfs25 are genetically linked and synergistic as *falciparum* malaria transmission-blocking vaccines. *Infect Immun* 65: <https://doi.org/10.1128/iai.65.3.1109-1113.1997>
- 7) Stowers AW, Keister DB, Muratova O, Kaslow DC. 2000. A Region of *Plasmodium falciparum* Antigen Pfs25 That Is the Target of Highly Potent Transmission-Blocking Antibodies. *Infect Immun* 68: <https://doi.org/10.1128/iai.68.10.5530-5538.2000>
- 8) de Graaf H, Payne RO, Taylor I, Miura K, Long CA, Elias SC, Zaric M, Minassian AM, Silk SE, Li L, Poulton ID, Baker M, Draper SJ, Gbesemete D, Brendish NJ, Martins F, Marini A, Mekhaïel D, Edwards NJ, Roberts R, Vekemans J, Moyle S, Faust SN, Berrie E, Lawrie AM, Hill F, Hill AVS, Biswas S. Safety and Immunogenicity of ChAd63/MVA Pfs25-IMX313 in a Phase I First-in-Human Trial. *Front Immunol*. 2021 Jul 14;12:694759. doi: 10.3389/fimmu.2021.694759. PMID: 34335606; PMCID: PMC8318801.
- 9) Werling, K., et al. (2019). "Steroid Hormone Function Controls Non-competitive *Plasmodium* Development in *Anopheles*." *Cell* 177(2): 315-325 e314.
- 10) Stryapunina I, Itoe MA, Trinh Q, Vidoudez C, Du E, Mendoza L, Hulai O, Kauffman J, Carew J, Shaw WR, Catteruccia F. Precise coordination between nutrient transporters ensures fertility in the malaria mosquito *Anopheles gambiae*. *PLoS Genet*. 2024 Jan 29;20(1):e1011145. doi: 10.1371/journal.pgen.1011145. PMID: 38285728; PMCID: PMC10852252.

Please see our response to each comment below.

Reviewer #2 (Remarks to the Author):

The authors' responses are acceptable. This reviewer agrees that understanding mosquito/parasite biology is important by itself. If the idea moving forward is develop a transmission blocking vaccine then it is important to understand the comparative TRA. The inclusion of the supplemental figure showing the comparative Pfs25 TRA is helpful. It should be understood that additional work will be required for mosGILT vaccine development but this provides a baseline.

We agree that additional work will be required for mosGILT vaccine development. A supplemental figure showing the comparative TRA of mosGILT antibodies to Pfs25 monoclonal antibodies has been included. (Fig. S7).

Reviewer #3 (Remarks to the Author):

The authors have addressed the majority of our previous concerns.

One of the previous comments remains unsolved, namely the question of whether prevalence (the percent of infected mosquitoes) and/or parasite infection intensity (the number of parasites within a infected mosquito, also referred to as parasite load) is reduced, as the authors claim throughout the manuscript.

The supplementary Datafile 3 shows that prevalence is reduced, which is the goal of any intervention strategy as a very small number of sporozoites is required to establish human infection.

However, in the experimental *Plasmodium falciparum* infections, parasite numbers per infected mosquito (those that have one or more parasites) are not reduced (Figure 2A, B). Thus part of the title of Fig. 2, namely "[...] decrease *P. falciparum* oocyst numbers[...]" is misleading, and so are statements throughout the manuscript that claim that MosGilt significantly reduces both oocyst infection intensity and prevalence.

The text should be modified to clarify the results, namely that prevalence is reduced.

The title of Fig. 2 now reads, "mosGILT antibodies decrease *P. falciparum* and *P. berghei* oocyst infection prevalence in a membrane-feeding model."

Reviewer #4 (Remarks to the Author):
